# LEARNING ENERGY-BASED MODELS BY SELF-NORMALISING THE LIKELIHOOD

## ABSTRACT

Training an energy-based model (EBM) with maximum likelihood is challenging due to the intractable normalisation constant. Traditional methods rely on expensive Markov chain Monte Carlo (MCMC) sampling to estimate the gradient of logartihm of the normalisation constant. We propose a novel objective called self-normalised log-likelihood (SNL) that introduces a single additional learnable parameter representing the normalisation constant compared to the regular log-likelihood. SNL is a lower bound of the log-likelihood, and its optimum corresponds to both the maximum likelihood estimate of the model parameters and the normalisation constant. We show that the SNL objective is concave in the model parameters for exponential family distributions. Unlike the regular log-likelihood, the SNL can be directly optimised using stochastic gradient techniques by sampling from a crude proposal distribution. We validate the effectiveness of our proposed method on various density estimation tasks as well as EBMs for regression. Our results show that the proposed method, while simpler to implement and tune, outperforms existing techniques.

## 1 INTRODUCTION

Energy-based models (EBMs) specify a probability density over a space $\mathcal{X}$ through a parameterised energy function $E_\theta : \mathcal{X} \to \mathbb{R}$. The associated density is then

$$p_\theta(x) = \frac{e^{-E_\theta(x)}}{Z_\theta}, \tag{1}$$

where $Z_\theta = \int e^{-E_\theta}(x)\,\mathrm{d}x$ is called the partition function or the normalising constant. However, $Z_\theta$ is often unknown and intractable, which makes training an EBM through maximum likelihood challenging.

Initial methods addresses the challenge with a pseudo-likelihood function, an altered version of the likelihood function that circumvents the need to compute the normalising constant (Besag, 1975; Mardia et al., 2009; Varin et al., 2011). Alternatively, gradients of the log-likelihood function can be estimated using the Boltzmann learning rule (Hinton and Sejnowski, 1983) or approximated using contrastive divergence (Hinton, 2002) at the price of expensive and difficult-to-tune Markov chain Monte Carlo (MCMC) sampling methods (Dalalyan, 2017; Welling and Teh, 2011). To alleviate this difficulty, Du and Mordatch (2019) proposed to maintain a buffer of samples during training using Langevin MCMC. This work was extended in Du et al. (2021) by considering a Kullback-Leibler divergence term that was claimed negligible in Hinton (2002). On another note, Xie et al. (2021) uses a flow trained alongside the EBM as a starting point for a short-term MCMC sampler reducing the dependency on long chains. In another work, Gao et al. (2021) proposed to train a succession of EBM on data diffused with noise allowing to train and sample on conditional distribution. Nijkamp et al. (2019) study the training of EBM for short-term

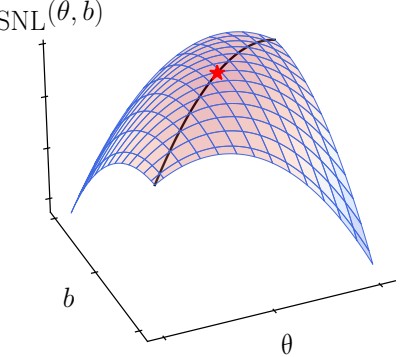

$\ell_{\mathrm{SNL}}(\theta, b)$

Figure 1: The SNL for a Gaussian with unknown mean $\theta \in \mathbb{R}$ and unit variance. The SNL a function of both $\theta$ and the additional parameter $b$, estimating the normalising constant. The black line corresponds to maximising $b$ for each given $\theta$, which exactly recovers the log-likelihood. The red star is the maximum log-likelihood, that is also the maximum of $\ell_{\mathrm{SNL}}(\theta, b)$, see details in Appendix B.

non-convergent Langevin Markov chain and showed excellent generation but was not optimizing the likelihood anymore. As it is critical to have a good estimate of this gradient, alternative methods consider using a proposal distribution $q$ together with importance sampling (Bengio and Senécal, 2003). However, this results in an objective that is an upper bound of the log-likelihood. Additionally, the choice of a proposal is critical, and a poor choice will lead to a loose bound. To tighten it, Geng et al. (2021) train the proposal to minimise the bound. This results in a min-max objective, similar to generative adversarial networks (GANs), which are infamous for their instability in training (Kumar et al., 2019; Farnia and Ozdaglar, 2021).

Another line of work aims at getting rid of the partition function altogether. It notably includes score matching and its variants (Hyvärinen, 2005; Vincent, 2011). Score matching is a family of objectives that circumvents the normalising constant by matching the Stein score of the data distribution (Stein, 1972) to the one of the model. Then different approaches present alternatives: implicit score matching trades the Stein score of the data distribution for the Hessian of the model (Hyvärinen, 2005; Kingma and Le Cun, 2010; Martens et al., 2012), while denoising score matching models instead a corrupted version of the data, which has a tractable density. The latter approach has proven to be very successful in generating high dimensional data, such as images and videos (Song et al., 2021; Ho et al., 2022). Another approach that bypasses the normalising constant is to minimise the Stein discrepancy (Barp et al., 2019; Grathwohl et al., 2020).

An alternative approach more related to our work is noise contrastive estimation (NCE), where Gutmann and Hyvärinen (2010) frames the problem as a logistic regression task between the data and a tractable noise distribution. This leads to a consistent estimate of the model parameters. Additionally, the normalisation constant is learned using an additional parameter (Mnih and Teh, 2012). The crucial issue of NCE, and that will not affect our method, is that the objective depends on the noise distribution, which is very hard to optimise (Chehab et al., 2022).

## 1.1 CONTRIBUTIONS

Our work is inspired by two papers on local likelihood density estimation (Loader, 1996; Hjort and Jones, 1996), which mention ways of bypassing the normalising constants in their quite specialised context. Our contributions are the following:

- We propose a new objective, the self-normalised log-likelihood (SNL) that is amenable to stochastic optimisation and allows to recover both the maximum likelihood estimate and its normalising constant;
- We study theoretical properties of the SNL, in particular its concavity for exponential families and its links with information geometry;
- We show on various low-dimensional tasks that SNL is straightforward to implement, and works as well or better than other, more complex, techniques for learning EBMs.
- We show state-of-the-art result on image regression dataset using an Energy Based Model.
- We derive a surrogate training objective, SNELBO, for a VAE with an EBM prior, that we train on binary MNIST.

## 2 SELF-NORMALISING THE LIKELIHOOD

We deal with some data $x_1, \ldots, x_n \in \mathcal{X}$, assumed to be independent and identically distributed samples from a distribution $p_{\text{data}}$. Our goal is to fit an EBM $p_\theta$, as defined in Eq. (1), to these data. The standard approach for fitting a probabilistic model is to maximise the likelihood function

$$\ell(\theta) = \frac{1}{n} \sum_{i=1}^{n} \log p_\theta(x_i). \tag{2}$$

Unfortunately, as we will discuss now, maximising such a function for an EBM is a daunting task.

## 2.1 WHY MAXIMUM LIKELIHOOD FOR EBMS IS HARD

Let us focus on a single data point $x$. The log density of our EBM is

$$\log p_\theta(x) = -E_\theta(x) - \log Z_\theta, \tag{3}$$

with $\theta$ being the learnable parameters of the model. Gradient-based methods are a popular approach to train an EBM via maximum likelihood; those methods require the gradient of the log density with respect to the parameters, $\theta$, that is

$$\nabla_\theta \log p_\theta(x) = -\nabla_\theta E_\theta(x) - \nabla_\theta \log Z_\theta. \tag{4}$$

While automatic differentiation can, usually, easily compute the gradient of the energy, $\nabla_\theta E_\theta(x)$, it is not the case for $\nabla_\theta \log Z_\theta$. However, following the Boltzmann learning rule (Hinton and Sejnowski, 1983), we can express the gradient of the normalising constant as an expected value (see e.g. Song and Kingma, 2021 for a full derivation):

$$\nabla_\theta \log Z_\theta = -\mathbb{E}_{X \sim p_\theta}[\nabla_\theta E_\theta(X)]. \tag{5}$$

We can obtain a Monte Carlo estimate of this gradient, but this requires sampling from the EBM itself, which leads to the use of MCMC-based methods that often suffer from poor stability and high computational cost. These procedures usually require very long chains to converge to the true distribution $p_\theta$. For the EBM to be computationally trainable, one needs to cut short the procedure. The obtained samples do not follow exactly $p_\theta$ meaning that the estimates of $\nabla_\theta \log Z_\theta$ are biased. As it is critical to have a good and fast estimate of this gradient, alternative methods consider using a proposal distribution $q$ in an importance sampling fashion, to yield a cheaper estimate:

$$\log Z_\theta = \log \int e^{-E_\theta(x)}\,\mathrm{d}x = \log \int \frac{e^{-E_\theta(x)}}{q(x)} q(x)\,\mathrm{d}x \geq \mathbb{E}_{X_1,\ldots,X_M \sim q}\left[\log \frac{1}{M}\sum_{m=1}^{M}\frac{e^{-E_\theta(X_m)}}{q(X_m)}\right], \tag{6}$$

where the last inequality is a consequence of Jensen's. In turn, this means that we will maximise the likelihood upper bound

$$\ell_{\mathrm{IS}}(\theta) = \frac{1}{n}\sum_{i=1}^{n} -E_\theta(x_i) - \mathbb{E}_{X_1,\ldots,X_M \sim q}\left[\log \frac{1}{M}\sum_{m=1}^{M}\frac{e^{-E_\theta(X_m)}}{q(X_m)}\right] \geq \ell(\theta), \tag{7}$$

in lieu of the likelihood. Depending on the choice of $q$, and on the number of importance samples $M$, this inequality is potentially very loose, meaning that one would train the model to maximise a biased approximation of the likelihood. Finding a good proposal $q$ that allows for fast sampling and correct estimation of its entropy is still a very active research area (Grathwohl et al., 2021; Kumar et al., 2019; Xie et al., 2018). Usually, this proposal is trained in parallel with the model $E_\theta$ which leads to a very unstable adversarial objective (Geng et al., 2021).

## 2.2 CAN WE MAKE THIS LOGARITHM DISAPPEAR?

The looseness of the importance sampling approximation $\ell_{\mathrm{IS}}(\theta)$ is only due to Jensen's inequality: if the logarithm were replaced by a linear function, it would be possible to compute unbiased estimated of the log-likelihood gradients. Our key idea is therefore to linearise the logarithm, using the following simple variational formulation. This will help us bypass the issues mentioned in Section 2.1.

**Lemma 2.1.** *For all $z > 0$,*

$$\log z = \min_{\lambda \in \mathbb{R}}\left(ze^{-\lambda} + \lambda - 1\right). \tag{8}$$

The proof of this lemma is elementary and provided in Appendix A.1. This result is often used as an illustration of variational representations in variational inference tutorials (see e.g. Jordan et al., 1999, Section 4.1; Ormerod and Wand, 2010, Section 3), but we are not aware of it being used in a context similar to ours. Applying Lemma 2.1 to Eq. (3) give us, for any $x \in \mathcal{X}$,

$$\begin{aligned}
\log p_\theta(x) &= -E_\theta(x) - \log Z_\theta = -E_\theta(x) - \min_{b \in \mathbb{R}}\left(e^{-b}Z_\theta + b - 1\right) \\
&= -E_\theta(x) + \max_{b \in \mathbb{R}}\left(-e^{-b}Z_\theta - b + 1\right) = \max_{b \in \mathbb{R}}\left(-E_\theta(x) - b - e^{-b}Z_\theta + 1\right).
\end{aligned} \tag{9}$$

Using Eq. (9), we define a new objective named the **self-normalised log-likelihood (SNL)** $\ell_{\mathrm{SNL}}$ that is a function of the original parameter of the EBM $\theta$ and a single additional parameter $b \in \mathbb{R}$:

$$\ell_{\mathrm{SNL}}(b, \theta) = \frac{1}{n}\sum_{i=1}^{n} -E_\theta(x_i) - b - e^{-b}Z_\theta + 1. \tag{10}$$

When maximised w.r.t. $b$, we can recover the exact log-likelihood of a given model $p_\theta$ and maximising both $\theta$ and $b$ leads to the maximum log-likelihood estimate, as formalised below.

**Theorem 2.1.** *For any given $\theta$, when the SNL is maximised with respect to $b$, we have access to the exact log-likelihood of the model.*

$$\max_{b \in \mathbb{R}} \ell_{\text{SNL}}(\theta, b) = \ell(\theta), \tag{11}$$

*Moreover, at the optimum, $b$ is the normalisation constant:*

$$\arg\max_{b \in \mathbb{R}} \ell_{\text{SNL}}(\theta, b) = \log Z_\theta. \tag{12}$$

*Finally, there is a one-to-one correspondence between the local optima of the SNL and the log-likelihood.*

The proof is available in Appendix A.2 and is a simple application of the variational formulation of the logarithm. The important consequence of this result is that *maximising the SNL w.r.t. $\theta$ and $b$ will recover both the maximum log-likelihood estimate and its normalising constant*. This ability of our objective to learn both the model and its normaliser motivates the name **self-normalised log-likelihood**. We chose to call the extra parameter $b$ because, when $E_\theta$ is model as a neural network, $b$ can simply be understood as the bias of its last layer.

Another direct consequence of Eq. (9) is that, for any $\theta$ and $b$, SNL is a lower bound of the log-likelihood. Using the importance-sampling upper bound, this will lead to useful "sandwichings" of the log-likelihood:

$$\ell_{\text{SNL}}(\theta, b) \leq \ell(\theta) \leq \ell_{\text{IS}}(\theta). \tag{13}$$

## 2.3 WHY MAXIMISING THE SNL IS EASIER

Why is the SNL more tractable than the standard log-likelihood? After all, the SNL also involves the intractable normalising constant. The key difference is that, since it depends linearly on it, it is now possible to obtain unbiased estimates of the SNL gradients.

Indeed, using a proposal $q$ gives us estimates of the gradient of $Z_\theta$ with importance sampling. Using

$$Z_\theta = \int \frac{e^{-E_\theta(x)}}{q(x)} q(x) dx = \mathbb{E}_{X \sim q} \left[ \frac{e^{-E_\theta(X)}}{q(X)} \right], \tag{14}$$

allows to get unbiased estimates of the SNL gradients w.r.t. $\theta$ and $b$. More precisely, for a batch of size $n_B$ and a number of samples $M$, we use the following estimate of the gradient w.r.t. $\theta$:

$$\nabla_\theta \ell_{\text{SNL}}(\theta, b) \approx -\frac{1}{n_b} \sum_{i=1}^{n_b} \nabla_\theta E_\theta(x_i) + e^{-b} \frac{1}{M} \sum_{m=1}^{M} \left[ \frac{\nabla_\theta E_\theta(x_m) e^{-E_\theta(x_m)}}{q(x_m)} \right]. \tag{15}$$

Similarly, we can compute unbiased estimates of the gradients w.r.t. $b$:

$$\nabla_b \ell_{\text{SNL}}(\theta, b) \approx -1 + e^{-b} \frac{1}{M} \sum_{m=1}^{M} \left[ \frac{e^{-E_\theta(x_m)}}{q(x_m)} \right]. \tag{16}$$

The theory of stochastic optimisation (see e.g. Bottou et al., 2018) then ensures that SGD-like algorithms, when applied to SNL, will converge to the maximum likelihood estimate and its normalising constant. In practice, we use popular algorithms like Adam to train $\theta$ and $b$ jointly. Some more specialised algorithms could also be used. For instance, Bietti and Mairal (2017) call optimisation problems similar to ours "infinite datasets with finite sum structure" (in our case, the infinite dataset are samples from the proposal, and the finite sum corresponds to the actual data), and propose an algorithm fit for this purpose. The full algorithm for training an energy based model with SNL is available in Appendix E with Algorithm 1.

**Are the gradients of $\ell$ and $\ell_{\mathrm{SNL}}$ related?** If we rewrite this gradient in the same fashion as Eq. (4), we can express the gradient of the SNL with the gradient of the log-likelihood :

$$\nabla_\theta \ell_{\mathrm{SNL}}(\theta, b) = -\frac{1}{n} \sum_{i=1}^{n} \nabla_\theta E_\theta(x) - e^{-b + \log Z_\theta} \nabla \log Z_\theta$$
$$= \nabla_\theta \ell_\theta + \nabla_\theta \log Z_\theta (1 - e^{-b + \log Z_\theta}). \tag{17}$$

When $b$ is equal to the normalisation constant $\log Z_\theta$, we can obtain an unbiased estimator of the true log-likelihood gradient.

## 2.4 PRACTICALITIES WHEN USING SNL

For EBMs to be well-posed, it is required that the normalisation constant exists, i.e. that $\int e^{-E_\theta(x)} dx < \infty$. To that end, following Grathwohl et al. (2020) and similarly to exponential tilting (Siegmund, 1976), multiplying the un-normalised probability by a density $d$ ensures the existence of the normalisation constant. The distribution becomes $p_\theta(x) \propto e^{-E_\theta(x)} d(x)$.

We call $d$ the base density or the base distribution. In the case where the proposal $q$ is equal to the base distribution, the SNL estimates and the gradient estimates simplify:

$$\nabla_\theta Z_\theta \approx \frac{1}{M} \sum_{m=1}^{M} \nabla_\theta E_\theta(x_m) e^{-E_\theta(x_m)}. \tag{18}$$

Furthermore, we initialise $b$ by estimating $\log Z_\theta$ with importance sampling using the proposal $q$ at the beginning of the training procedure. This practice allows us to get gradient estimates of SNL somewhat close to the true gradient log-likelihood.

## 2.5 RELATED WORKS

Objectives similar to SNL have been proposed in the past. In particular, in the context of local likelihood density estimation, Loader (1996) and Hjort and Jones (1996) handled intractable normalising constants in a similar fashion to ours. Arbel et al. (2020) leveraged a similar approach to estimate the normalisation constant in order to train hybrids of generative adversarial nets and EBMs. Neither of these works used importance sampling. Pihlaja et al. (2010) and Gutmann and Hirayama (2011) proposed families of generalisations of NCE which contain an objective similar to SNL as a special case. In these generalisations of NCE, the noise distribution plays a similar role to our proposal, but the obtained estimates in general differ from maximum likelihood. The novelty of SNL lies in the fact that it allows to perform exact maximum likelihood optimisation (regardless of the choice of proposal) for an EBM using stochastic optimisation together with importance sampling.

# 3 SOME THEORETICAL PROPERTIES OF SNL

## 3.1 CONCAVITY OF SNL FOR EXPONENTIAL FAMILIES

It is a well-known fact that the log-likelihood of exponential families is concave because of the particular form of the gradient of the normaliser. We provide a proof in Appendix A.3 for completeness. The self-normalised log-likelihood preserves this property with the exponential family: the SNL is even *jointly* concave in both parameters.

**Theorem 3.1.** *If $(p_\theta)_\theta$ is a canonical exponential family, then $\ell_{\mathrm{SNL}}(\theta, b)$ is jointly concave.*

The proof is available in Appendix A.4 and follows directly from the convexity of the exponential. This means that the many theoretical results on stochastic optimisation for convex functions could be leveraged to prove convergence guarantees of SNL (see e.g. Bottou et al., 2018).

## 3.2 AN INFORMATION-THEORETIC INTERPRETATION

Maximum likelihood has the following classical information-theoretic interpretation: when the number of samples goes to infinity, maximising the likelihood is equivalent to minimising the

Kullback-Leibler divergence between $p_\theta$ and the true data distribution $p_{\text{data}}$ (see e.g. White, 1982). A similar rationale exists also for SNL, and involves a generalisation of the Kullback-Leibler divergence to un-normalised finite measures. This generalisation exists also in the more general context of $f$-divergences, as detailed for instance by Amari and Nagaoka (2000, Section 3.6) or Stummer and Vajda (2010). It reduces to the usual definition when $f_1$ and $f_2$ are probability densities and shares many of the merits of the usual Kullback-Leibler divergence (see Appendix D for more details).

Standard maximum likelihood is asymptotically equivalent to minimising $\text{KL}(p_{\text{data}}||p_\theta)$. As we detail in Appendix D, this turns out to be equivalent to minimising the generalised divergence between $p_{\text{data}}$ and all un-normalised models proportional to $e^{-E_\theta}$:

$$\text{KL}(p_{\text{data}}||p_\theta) = \min_{c>0} \text{KL}(p_{\text{data}}||ce^{-E_\theta}). \tag{19}$$

This new divergence is related to the SNL in the same way that the standard Kullback-Leibler divergence is related to the likelihood. Indeed, for any $c > 0$,

$$\text{KL}(p_{\text{data}}||ce^{-E_\theta}) = \int \log\left(\frac{p_{\text{data}}(x)}{ce^{-E_\theta(x)}}\right) p_{\text{data}}(x)(x)dx + cZ_\theta - 1 \tag{20}$$

$$= -\int e^{-E_\theta(x)} p_{\text{data}}(x)dx - \log c + cZ_\theta - 1 + \underbrace{\int \log(p_{\text{data}}(x))p_{\text{data}}(x)dx}_{\text{does not depend on } \theta \text{ nor } c}. \tag{21}$$

The first integral, that depends on $\theta$, is intractable, but may be estimated if we have access to an i.i.d. dataset $x_1, \ldots, x_n$, leading to the estimate

$$\text{KL}(p_{\text{data}}||ce^{-E_\theta}) \approx -\frac{1}{n}\sum_{i=1}^n e^{-E_\theta(x_i)} - \log c + cZ_\theta - 1 + \int \log(p_{\text{data}}(x))p_{\text{data}}(x)\,\mathrm{d}x \tag{22}$$

$$= -\ell_{\text{SNL}}(\theta, \log c) + \int \log(p_{\text{data}}(x))p_{\text{data}}(x)\,\mathrm{d}x, \tag{23}$$

which means that minimising the SNL will asymptotically resemble minimising the generalised Kullback-Leibler divergence. In the context of local likelihood density estimation, Hjort and Jones (1996) also derived similar connections with the generalised Kullback-Leibler divergence. More recently, Bach (2022) applied the same variational representation of the logarithm to the generalised Kullbakc-Leibler, in a context very different from ours.

## 4 Extending SNL

### 4.1 Self-normalization in the regression setting

We consider the supervised regression problem where we are given a dataset of pairs of inputs and targets $(x, y) \in \mathcal{X} \times \mathcal{Y}$ where the target space $\mathcal{Y}$ is continuous. We want to estimate the conditional distribution $p_{\text{data}}(y|x)$ using an EBM:

$$p_\theta(y|x) = \frac{e^{-E_\theta(x,y)}}{Z_{\theta,x}}, \tag{24}$$

where $Z_{\theta,x} = \int e^{-E_\theta(x,y)}\,\mathrm{d}y$. The main difference with the previous density estimation setup is that the normalisation constant $Z_{\theta,x}$ also depends on the input value $x$.

Because the normaliser now also depends on $x$, we introduce a new family of functions $b_\phi$ whose role is to estimate the normalisation constant $Z_{\theta,x}$. Similarly to the density estimation case, we define the self-normalised log-likelihood as

$$\ell_{\text{SNL}}(\theta, \phi) = \frac{1}{n}\sum_{i=1}^n \left(-E_\theta(x_i, y_i) - b_\phi(x_i) - Z_{\theta,x_i}e^{-b_\phi(x_i)} + 1\right). \tag{25}$$

Provided the family $b_\phi$ is expressive enough, this SNL for regression enjoys the same properties as its unsupervised counterpart. We can retrieve the maximum likelihood estimate when maximising

| Objective | Base Dist | Funnel | | Pinwheel | | Checkerboard | | Four Circles | |
|---|---|---|---|---|---|---|---|---|---|
| | | $\ell_{\text{IS}}$ | $\ell_{\text{SNL}}$ | $\ell_{\text{IS}}$ | $\ell_{\text{SNL}}$ | $\ell_{\text{IS}}$ | $\ell_{\text{SNL}}$ | $\ell_{\text{IS}}$ | $\ell_{\text{SNL}}$ |
| NCE | $\mathcal{N}(0,1)$ | $-2.040_{(\pm 0.251)}$ | $-2.044_{(\pm 0.254)}$ | $\mathbf{-1.004}_{(\pm 0.072)}$ | $\mathbf{-1.020}_{(\pm 0.084)}$ | $-1.947_{(\pm 0.033)}$ | $-1.964_{(\pm 0.032)}$ | $-2.117_{(\pm 0.005)}$ | $-2.120_{(\pm 0.006)}$ |
| SNL | $\mathcal{N}(0,1)$ | $\mathbf{-1.811}_{(\pm 0.175)}$ | $\mathbf{-1.831}_{(\pm 0.181)}$ | $-1.031_{(\pm 0.066)}$ | $-1.035_{(\pm 0.065)}$ | $\mathbf{-1.902}_{(\pm 0.012)}$ | $\mathbf{-1.905}_{(\pm 0.012)}$ | $\mathbf{-1.914}_{(\pm 0.022)}$ | $\mathbf{-1.918}_{(\pm 0.022)}$ |
| NCE | None | $-1.894_{(\pm 0.096)}$ | $-1.896_{(\pm 0.097)}$ | $-1.063_{(\pm 0.019)}$ | $-1.069_{(\pm 0.024)}$ | $-1.997_{(\pm 0.022)}$ | $-2.025_{(\pm 0.056)}$ | $-2.231_{(\pm 0.038)}$ | $-2.232_{(\pm 0.039)}$ |
| SNL | None | $-2.006_{(\pm 0.378)}$ | $-2.066_{(\pm 0.468)}$ | $-1.072_{(\pm 0.040)}$ | $-1.086_{(\pm 0.030)}$ | $-1.966_{(\pm 0.030)}$ | $-1.969_{(\pm 0.028)}$ | $-1.971_{(\pm 0.047)}$ | $-1.973_{(\pm 0.048)}$ |

Table 1: Evaluation of the performance of EBMs trained with NCE or SNL objective with or without a base distribution. We generate each datasets five times and run each set of parameter once on each. We report the mean and standard deviation of the estimated log-likelihood and the self-normalised likelihood $\ell_{SNL}$. Highest is best.

the SNL in both $\theta$ and $\phi$. Moreover, at the optimum, for any $x \in \mathcal{X}$, $b_\phi(x)$ is the normalisation constant $\log Z_{\theta,x}$. The SNL for regression is also a lower bound of the true conditional log-likelihood. Following reasoning of Section 2.3, we propose to train an EBM model for regression using the SNL. To that end, we consider a proposal $q_\psi$ that depends on both $x$ and $y$ and is parameterised by $\psi$. For instance, Gustafsson et al. (2020) use a mixture density network (MDN, Bishop, 1994) proposal. In Gustafsson et al. (2020), the EBM is trained jointly with the MDN. The MDN maximisation objective is an average combination between the negative Kullback-Leibler divergence between the $p_\theta$ and $q_\psi$.

### 4.2 SELF-NORMALISED EVIDENCE LOWER BOUND

The SNL aproach allows training a variational auto-encoder (VAE) with an energy-based prior using approximate inference. Both Pang et al. (2020) and Schröder et al. (2023) trained an EBM as a prior in the latent space for a noisy sampler but required MCMC to sample from the posterior and the prior during training. We introduce the self-normalised evidence lower bound (SNELBO), a surrogate ELBO objective that leverages the self-normalised log-likelihood to allow for straightforward training.

Formally, we consider a variational auto-encoder Kingma and Welling (2014) with a prior $p_\theta(z)$ defined by an EBM composed of an energy function $E_\theta$ parameterised by a neural network and an associated base distribution $d(z)$, i.e., $p_\theta(z) = \frac{e^{-E_\theta}d(z)}{Z_\theta}$ where $Z_\theta = \int e^{-E_\theta(z)}d(z)dz$. The generative model is the same as in VAE and an output density $p_\phi(z|x)$ is parameterised by a neural network $g_\phi(z)$. Since the likelihood is intractable, we posit a conditional variational distribution $q_\gamma(z|x)$ to approximate the posterior of the model, similarly to the original VAE. Using Lemma 2.1, we can obtain the SNELBO:

$$\mathcal{L}_{\text{SNL}}(\theta, \phi, \gamma, b) = \mathbb{E}_{q_\gamma(z|x)}\left[\log p_\phi(x|z)\right] + \mathbb{E}_{q_\gamma(z|x)}\left[\log \frac{d(z)}{q_\gamma(z|x)}\right]$$
$$+ \mathbb{E}_{q_\gamma(z|x)}\left[-E_\theta(z) - b\right] - \mathbb{E}_{d(z)}\left[e^{-E_\theta(z)-b}\right] + 1. \quad (26)$$

We note that the SNELBO is a lower bound on the log-likelihood, $\ell(\theta, \phi)$, and a lower bound on the regular ELBO, $\mathcal{L}$, that is tight for optimal $b$, i.e., $\ell(\theta, \phi) \geq \mathcal{L}(\theta, \phi, \gamma) \geq \mathcal{L}_{\text{SNL}}(\theta, \phi, \gamma, b)$ and $\mathcal{L}(\theta, \phi, \gamma) = \max_{b \in \mathbb{R}} \mathcal{L}_{\text{SNL}}(\theta, \phi, \gamma, b)$. See Appendix F for derivation details. This surrogate objective can be interpreted as the combination of the ELBO from a VAE whose prior is the base distribution $d(z)$ with a regularization term from the EBM. As such, the EBM can be added easily on top of any VAE model.

## 5 EXPERIMENTS

### 5.1 DENSITY ESTIMATION

We evaluate the performances of an EBM on density estimation, trained with SNL on four different, two-dimensional, generated datasets. Each dataset has 7000 samples in its training set, 2000 samples in its test set and 1000 in its validation set. Each model is trained with the Adam optimiser Kingma and Ba (2015), with a learning rate of $1e-3$, for 25 epochs and with a standard Gaussian proposal. We compare our model with an EBM trained in the same condition but with noise contrastive estimation. Both setups uses a fully connected network architecture specified in Table 8. Both setup also leverages a base distribution that equals the proposal distribution and that is not trained. Fig. 4 shows our

| | EBM - SNL | Gaussian | MADE | MADE MoG | Real NVP (5) | Real NVP (10) | MAF (5) | MAF (10) | MAF MoG (5) |
|---|---|---|---|---|---|---|---|---|---|
| Power ($d = 6$) | $[0.28, 0.41]$ | $-7.74$ | $-3.08$ | $0.40$ | $-0.02$ | $0.17$ | $0.14$ | $0.24$ | $0.30$ |
| Gas ($d = 8$) | $[5.73, 7.74]$ | $-3.58$ | $3.56$ | $8.47$ | $4.78$ | $8.33$ | $9.07$ | $10.08$ | $9.59$ |
| Hepmass ($d = 21$) | $[-19.22, -19.20]$ | $-27.93$ | $-20.98$ | $-15.15$ | $-19.62$ | $-18.71$ | $-17.70$ | $-17.73$ | $-17.39$ |

Table 2: For EBM-SNL the upper bound corresponds to $\ell_{IS}$ and the lower bound to $\ell_{SNL}$. Both are computed using 20000 samples from the test set.

results. Qualitatively, we observe that the two models perform on par, except for the four circles dataset, where SNL dominates.

We perform a second set of density estimation experiments, with the same setup as above, and explore the impact of the base distribution. For that purpose we perform a set of experiment with a proposal and a set without. We train each configuration five times, re-generating the dataset every time. We show our results in Table 1. According to those results, SNL-trained EBM with a base distribution, perform better than their NCE counter part across all datasets except Pinwheel. Contrary to an SNL trained EBM, we observe that using a base distribution with a NCE trained EBM is detrimental to its performances.

We experiment our SNL-trained EBM on the UCI datasets, to explore the impact of higher dimensions. We use a simple gaussian with full covariance as the proposal. We report our results in Table 2, where we beat or are on par with other method; and, to the best of our knowledge, it is the first competitive application of EBMs in this setting.

## 5.2 EBMs FOR REGRESSION

Following Gustafsson et al. (2022), we study and compare our training method on two one-dimensional regression tasks (see Figure Fig. 3 in Appendix G) and four image regression datasets. We parameterise our model with the same architecture as Gustafsson et al. (2022) where the output of a feature extractor, $h_x$, feeds both the proposal $q_\psi(.|h_x)$ and a head neural network for the EBM (see Figure 1 in Gustafsson et al. (2022) for more details). With that formulation, the feature extractor is learned only with gradients calculated from the EBM loss (either NCE or SNL in our case) and the proposal is learned with a fixed feature extractor.

In our experiments, we consider three different proposals:

- A mixture density network proposal whose parameters are given by a small fully connected neural network.

- A fixed multivariate Gaussian $\mathcal{N}(\mu, \Sigma)$ whose parameters are estimated before training with the training dataset and fixed during training.

- A fixed uniform distribution $\mathcal{U}$ that is defined by leveraging the knowledge from the dataset and fixed during training.

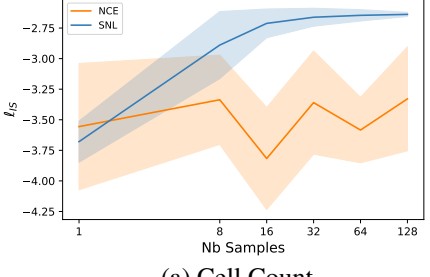

(a) Cell Count

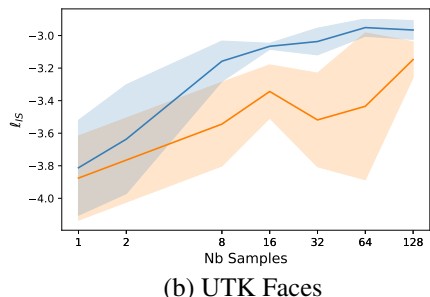

(b) UTK Faces

Figure 2: Performance evolution of the EBMs for regression trained on Cell Count and UTKFaces dataset.

All models are evaluated using an estimate of the log-likelihood with $M = 20,000$ samples from a multivariate Gaussian whose parameters are estimated before training:

$$\ell_{IS} = \frac{1}{N} \sum_{i=1}^{N} \left( -E_\theta(x_i, y_i) - b_\phi(x_i) - \log \frac{1}{M} \sum_{m=1}^{M} e^{-E_\theta(x_i, y_i^{(m)}) - b_\phi(x_i)} \right) \quad (27)$$

Since NCE normalises the EBM at the optimum (Mnih and Teh, 2012), we also provide the $\ell_{\text{SNL}}$ (i.e. a lower bound estimate of the log-likelihood) for each set of parameters using the same proposal

| Objective | Proposal $q$ | Regression Dataset 1 | | Regression Dataset 2 | |
|---|---|---|---|---|---|
| | | $\ell_{\text{IS}}$ | $\ell_{\text{SNL}}$ | $\ell_{\text{IS}}$ | $\ell_{\text{SNL}}$ |
| NCE | $\mathcal{N}(\mu,\Sigma)$ | $-0.030$ $_{(\pm0.278)}$ | $-0.718$ $_{(\pm0.256)}$ | $-2.592$ $_{(\pm0.214)}$ | $-3.559$ $_{(\pm1.881)}$ |
| NCE | MDN K2 | $-0.611$ $_{(\pm0.154)}$ | $-1.492$ $_{(\pm0.993)}$ | $-2.451$ $_{(\pm0.088)}$ | $-2.634$ $_{(\pm0.084)}$ |
| SNL | $\mathcal{N}(\mu,\Sigma)$ | $0.164$ $_{(\pm0.088)}$ | $0.033$ $_{(\pm0.077)}$ | $\mathbf{-1.813}$ $_{(\pm0.109)}$ | $\mathbf{-1.836}$ $_{(\pm0.109)}$ |
| SNL | MDN K2 | $\mathbf{0.255}$ $_{(\pm0.017)}$ | $\mathbf{0.251}$ $_{(\pm0.016)}$ | $-2.099$ $_{(\pm0.250)}$ | $-2.170$ $_{(\pm0.353)}$ |

Table 3: Evaluation of regression EBMs on the 1D toy regression problems with two different objectives and two different proposals. Each model is trained for five runs and we report the mean and standard deviation of the estimated log-likelihood $\ell_{\text{IS}}$ and the self normalized log-likelihood $\ell_{\text{SNL}}$. Using the SNL as objective clearly outperforms the NCE.

| Objective | Proposal | Steering Angle | | Cell Count | | UTKFaces | | BIWI | |
|---|---|---|---|---|---|---|---|---|---|
| | | $\ell_{\text{IS}}$ | $\ell_{\text{SNL}}$ | $\ell_{\text{IS}}$ | $\ell_{\text{SNL}}$ | $\ell_{\text{IS}}$ | $\ell_{\text{SNL}}$ | $\ell_{\text{IS}}$ | $\ell_{\text{SNL}}$ |
| NCE | $\mathcal{N}(\mu,\Sigma)$ | $-3.649$ $_{(\pm1.224)}$ | UNNORMALIZED | $-3.367$ $_{(\pm0.399)}$ | $-9.675$ $_{(\pm0.605)}$ | $-3.147$ $_{(\pm0.1100)}$ | $-8.223$ $_{(\pm3.795)}$ | $-11.02$ $_{(\pm0.576)}$ | UNNORMALIZED |
| NCE | MDN-8 | $-4.001$ $_{(\pm0.667)}$ | UNNORMALIZED | $-3.864$ $_{(\pm0.048)}$ | UNNORMALIZED | $-4.123$ $_{(\pm0.21)}$ | $-5.170$ $_{(\pm0.955)}$ | $-11.998$ $_{(\pm0.339)}$ | UNNORMALIZED |
| SNL | $\mathcal{N}(\mu,\Sigma)$ | $-2.665$ $_{(\pm1.37)}$ | $-3.973$ $_{(\pm3.15)}$ | $-2.701$ $_{(\pm0.041)}$ | $-2.725$ $_{(\pm0.046)}$ | $-2.966$ $_{(\pm0.057)}$ | $-2.991$ $_{(\pm0.069)}$ | $-10.86$ $_{(\pm1.017)}$ | $-11.05$ $_{(\pm1.141)}$ |
| SNL | Uniform | $\mathbf{-1.402}$ $_{(\pm0.068)}$ | $\mathbf{-1.423}$ $_{(\pm0.074)}$ | $\mathbf{-2.604}$ $_{(\pm0.001)}$ | $\mathbf{-2.620}$ $_{(\pm0.007)}$ | $-2.927$ $_{(\pm0.032)}$ | $\mathbf{-2.965}$ $_{(\pm0.019)}$ | $-10.44$ $_{(\pm0.138)}$ | $-10.51$ $_{(\pm1.222)}$ |
| SNL | MDN-8 | $-1.673$ $_{(\pm0.042)}$ | $-1.692$ $_{(\pm0.046)}$ | $-2.801$ $_{(\pm0.071)}$ | $-2.811$ $_{(\pm0.071)}$ | $\mathbf{-2.921}$ $_{(\pm0.055)}$ | $-2.943$ $_{(\pm0.062)}$ | $\mathbf{-10.01}$ $_{(\pm0.092)}$ | $\mathbf{-10.04}$ $_{(\pm0.091)}$ |

Table 4: Evaluation of EBMs for regression on image regression datasets with two different objectives and different proposals. Each model is trained for five runs and we report the mean and standard deviation of the estimated log-likelihood ($\ell_{IS}$) and estimated self-normalized log-likelihood ($\ell_{SNL}$). When the proposal is MDN, the proposal is learned jointly with the model following Gustafsson et al. (2022).

as $\ell_{\text{IS}}$ with 20000 samples. If $\ell_{\text{SNL}}$ is close to $\ell_{\text{IS}}$, this means that the lower-bound is tight and $b_\phi$ approximates correctly $\log Z_\theta$ (or if no $b_\phi$ is used the network is self-normalised and $Z_\theta = 1$).

### 5.2.1 1D REGRESSION DATASETS

We consider here the two artificial datasets for 1D regression with multimodal distribution $p(y|x)$ (see Fig. 3). We provide a description of the neural network architecture in Appendix I.2. On both datasets, the SNL always outperformed its NCE counterparts with respect to the estimated upper bound $\ell_{\text{IS}}$ Table 3. Moreover, the $\ell_{\text{SNL}}$ of the NCE is loose compared to the $\ell_{\text{SNL}}$. We provide additional results in Table 6. Using a base distribution to ensure the existence of the normalisation constant $Z_\theta$ either improves or gives similar results with the SNL objective but systematically damages the results when minimising the NCE loss. As mentioned by Mnih and Teh (2012), with both objectives, explicitly modelling $b_\phi$ does not provide a better estimation of the network. The normalisation is implicitly learned with $E_\theta$.

### 5.2.2 IMAGE REGRESSION DATASETS

We train an NCE-EBM setup as well as an SNL-EBM setup on an image regression task. We train on four different datasets, steering angle, cell count, UTKFaces and BIWI and follow the same setup as Gustafsson et al. (2022). Similarly to the 1D regression datasets, SNL-trained EBM always outperforms its NCE counterparts Table 4. When using NCE, the resulting energy is often un-normalised whereas SNL trained EBM provides a tighter bound. In Fig. 2, we observe that our method improves with the number of samples but stagnates after $M = 64$ samples. On the other hand, NCE seems to improve with the sample size but in a less compelling fashion. We provide additional results in Table 7.

### 5.3 VAE WITH LATENT PRIOR EBM

We train a VAE with EBM prior on binary MNIST using SNELBO, as outlined in Section 4.2. We parameterise the output distribution with a Bernoulli distribution with parameters from a neural network $g_\phi$, i.e. $p_\phi(x|z) = \mathcal{B}(x|g_\phi(z))$ and the approximate posterior with a Gaussian whose parameters are given by a neural network $q_\gamma(z|x)$. We either train from scratch the VAE with EBM prior (VAE-EBM) or we only train the prior of a pre-trained VAE with standard Gaussian prior (VAE-EBM Post-hoc). We compare to a standard VAE with

| | |
|---|---|
| VAE | -89.10 |
| VAE-MoG | -88.73 |
| VAE-EBM Post-Hoc | -88.11 |
| VAE-EBM | **-87.09** |

Table 5: ELBO/SNELBO for VAEs with different priors.

Gaussian prior (VAE) and a VAE with a Mixture of Gaussian Prior (VAE-MoG) and 10 mixtures. All VAEs are trained with a latent space of size 16. In Table 5, we show that training VAE-EBM with latent EBM provides better SNELBO.

## 6 CONCLUSION

We proposed a new objective to train energy-based models (EBMs) called self-normalising log-likelihood (SNL). By maximising SNL with respect to the parameters of the EBM and an additional single parameter $b$, we can recover both the maximum likelihood estimate and the normalising constant at optimality. We conducted an extensive experimental study on low-dimension datasets for density estimation, complex regression problems and training VAEs with EBM prior.

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

# Supplementary material:
# Learning energy-based models
# by self-normalising the likelihood

## A  PROOFS

### A.1  VARIATIONAL log

Inspired by Jordan et al. (1999) and Ormerod and Wand (2010), we show the following lemma:

**Lemma 2.1.** *For all $z > 0$,*

$$\log z = \min_{\lambda \in \mathbb{R}} \left( ze^{-\lambda} + \lambda - 1 \right). \tag{8}$$

*Proof.* Let $z > 0$ and $\lambda \in \mathbb{R}$, we define the function:

$$h(\lambda) = ze^{-\lambda} + \lambda - 1. \tag{28}$$

By differentiating this function with respect to $\lambda$, we get:

$$h'(\lambda) = -ze^{-\lambda} + 1. \tag{29}$$

The differentiated function $h'$ is negative for $\lambda < \log z$ and positive for $\lambda > \log z$. Thus the minimum of $h$ is reached at $\lambda = \log(z)$ and $h(\log(z)) = \log(z)$, hence the proof. □

### A.2  PROOF AND THEOREM 2.1

We begin by reminding notations: we consider an energy-based model, which specifies a probability density over a space $\mathcal{X}$ through a parameterised energy function $E_\theta : \mathcal{X} \to \mathbb{R}$. The associated density is:

$$p_\theta(x) = \frac{e^{-E_\theta(x)}}{Z_\theta}, \tag{30}$$

where $Z_\theta = \int e^{-E_\theta}(x)\mathrm{d}x$ is the partition function. Given $n$ data points $x_1, ..., x_n \in \mathcal{X}$, we define the log-likelihood function:

$$\ell(\theta) = \frac{1}{n} \sum_{i=1}^{n} -E_\theta(x_i) - \log Z_\theta. \tag{31}$$

We define as well the self-normalised log-likelihood (SNL) as:

$$\ell_{\mathrm{SNL}}(\theta, b) = \frac{1}{n} \sum_{i=1}^{n} -E_\theta(x_i) - b - e^{-b} Z_\theta + 1. \tag{32}$$

We now recall Theorem 2.1:

**Theorem 2.1.** *For any given $\theta$, when the SNL is maximised with respect to $b$, we have access to the exact log-likelihood of the model.*

$$\max_{b \in \mathbb{R}} \ell_{\mathrm{SNL}}(\theta, b) = \ell(\theta), \tag{11}$$

*Moreover, at the optimum, $b$ is the normalisation constant:*

$$\arg\max_{b \in \mathbb{R}} \ell_{\mathrm{SNL}}(\theta, b) = \log Z_\theta. \tag{12}$$

*Finally, there is a one-to-one correspondence between the local optima of the SNL and the log-likelihood.*

*Proof.* Using Lemma 2.1, we show that for any $\theta$, $\max_{b \in \mathbb{R}} \ell_{\mathrm{SN}}(\theta, b) = \ell(\theta)$.

$$\ell(\theta) = \frac{1}{n} \sum_{i=1}^{n} \log p_\theta(x_i) \tag{33}$$

$$= \frac{1}{n} \sum_{i=1}^{n} \log(e^{-E_\theta(x_i)}) - \log Z_\theta \tag{34}$$

$$= \frac{1}{n} \sum_{i=1}^{n} \log(e^{-E_\theta(x_i)}) - \min_{b \in \mathbb{R}}(e^{-b} Z_\theta + b - 1) \tag{35}$$

$$= \frac{1}{n} \sum_{i=1}^{n} \log(e^{-E_\theta(x_i)}) + \max_{b \in \mathbb{R}}(-e^{-b} Z_\theta - b + 1) \tag{36}$$

$$= \max_{b} \frac{1}{n} \sum_{i=1}^{n} \log(e^{-E_\theta(x_i)}) - e^{-b} Z_\theta - b + 1 \tag{37}$$

$$= \max_{b \in \mathbb{R}} \ell_{\mathrm{SNL}}(\theta, b). \tag{38}$$

We show that $\ell(\theta)$ and $\ell_{SNL}(\theta, b)$ share the same local maxima.

Let $\theta^*$ a local optimum of $\ell(\theta^*)$, we will construct a local optimum of $\ell_{SNL}$.

Let $b^* = \log Z_{\theta^*}$, then the gradient of $\nabla_{\theta, b} \ell_{SNL} = 0$ :

$$\nabla_b \theta \ell_{SNL}(\theta^*, b)(x) = -1 + Z_{\theta^*} e^{-b} = 0 \tag{39}$$

$$\nabla_\theta \ell_{SNL}(\theta^*, b)(x) = -\nabla_\theta E_{\theta^*}(x) - e^{-b} \nabla_\theta Z_\theta \tag{40}$$

$$= -\nabla_\theta E_{\theta^*}(x) - e^{-b} Z_{\theta^*} \nabla_\theta \log Z_{\theta^*} \tag{41}$$

$$= -\nabla_\theta E_{\theta^*}(x) - \frac{1}{Z_{\theta^*}} Z_{\theta^*} \nabla_\theta \log Z_{\theta^*} \tag{42}$$

$$= -\nabla_\theta E_\theta(x) - \nabla_\theta \log Z_{\theta^*} \tag{43}$$

$$= \nabla_\theta \ell(\theta^*)(x) \tag{44}$$

$$= 0 \tag{45}$$

Thus, for any local optimum of $\ell(\theta^*)$, the pair $(\theta^*, \log Z_{\theta^*}$ is a local optimum $\ell_{SNL}$.

Conversely, with the same reasoning, for any pair of $(\tilde{\theta}, \tilde{b})$ local optimum of $\ell_{SNL}$, $\tilde{\theta}$ is a local maximum of $\ell$.

$\square$

### A.3 PROOF OF THEOREM A.1

For completeness, we begin by proving the classical result about the convexity of exponential families. For more details, see e.g. Wainwright and Jordan (2008, Chapter 3).

**Theorem A.1.** *If $(p_\theta)_\theta$ is a canonical exponential family, then $\ell(\theta)$ is concave. In particular, the gradient and the Hessian of $\log Z_\theta$ are respectively the mean and the covariance matrix of the sufficient statistics.*

*Proof.* Let's consider an exponential family whose densities with respect to a base measure are parameterised as $p_\theta(x) = e^{\theta^T s(x) - \log Z_\theta}$, where $s(x)$ is the sufficient statistics and $\theta$ the natural parameters. To simplify formulas, we will assume that we observe a single data point $x \in \mathcal{X}$.

Observing several i.i.d. data points will preserve concavity because a sum of concave functions remains concave, so there is no loss of generality.

The log-likelihood of such a model is given by:

$$\ell(\theta) = \theta^T s(x) - \log Z_\theta = \theta^T s(x) - \log \int e^{\theta^T s(x)} \mathrm{d}x. \tag{46}$$

We will prove that this objective is concave by showing that the Hessian is negative semi-definite. Let's calculate the gradient and Hessian of $\log Z_\theta$. For integrals akin to the normalising constant, switching differentiation and integration is allowed (see e.g. Lehmann and Romano, 2022, Theorem 2.7.1), and we get

$$\nabla_\theta \log Z_\theta = \nabla_\theta \log \int e^{\theta^T s(x)} dx \tag{47}$$

$$= \frac{\int s(x) e^{\theta^T s(x)} dx}{\int e^{\theta^T s(x)} dx} \tag{48}$$

$$= \int s(x) e^{\theta^T s(x) - \log Z_\theta} \mathrm{d}x \tag{49}$$

$$= \mathbb{E}_\theta[s(x)]; \tag{50}$$

$$H_\theta(\log Z_\theta) = \int s(x) s(x)^T e^{\theta^T s(x) - \log Z_\theta} \mathrm{d}x - \left( \int s(x) e^{\theta^T s(x) - \log Z_\theta} \mathrm{d}x \right) (\nabla_\theta \log Z_\theta)^T \tag{51}$$

$$= \int s(x) s(x)^T e^{\theta^T s(x) - \log Z_\theta} \mathrm{d}x \tag{52}$$

$$- \left( \int s(x) e^{\theta^T s(x) - \log Z_\theta} \mathrm{d}x \right) \left( \int s(x) e^{\theta^T s(x) - \log Z_\theta} \mathrm{d}x \right)^T \tag{53}$$

$$= \mathbb{E}_\theta[s(x) s(x)^T] - \mathbb{E}_\theta[s(x)] \mathbb{E}_\theta[s(x)]^T \tag{54}$$
$$= \mathbb{V}_\theta[s(x)]. \tag{55}$$

Using the Hessian of $\log Z_\theta$, we can express directly the Hessian of the log-likelihood $\ell(\theta)$:

$$H(\ell(\theta)) = -\mathbb{V}_\theta[s(x)]. \tag{56}$$

The covariance matrix $\mathbb{V}_\theta[s(x)]$ is positive semi-definite thus the hessian $H(\ell(\theta))$ is negative semi-definite. Hence, $\ell(\theta)$ is concave.

$\square$

### A.4 Proof of Theorem 3.1

**Theorem 3.1.** *If $(p_\theta)_\theta$ is a canonical exponential family, then $\ell_{\mathrm{SNL}}(\theta, b)$ is jointly concave.*

*Proof.* Using the same notations as the previous proof, our exponential family is parameterised as $p_\theta(x) = e^{\theta^T s(x) - \log Z_\theta}$, where $s(x)$ is the sufficient statistics and $\theta$ the natural parameters. We again assume without loss of generality that we observe a single data point $x \in \mathcal{X}$.

The self-normalised log-likelihood is as follows:

$$\ell_{\mathrm{SNL}}(\theta, b) = \theta^T s(x) - b - e^{-b + \log Z_\theta} + 1 \tag{57}$$

$$= \theta^T s(x) - b + 1 - \int e^{\theta^T s(x) - b} \mathrm{d}x. \tag{58}$$

Since the first term of the equation is affine, we will show that the function $(\theta, b) \mapsto e^{-b + \log Z_\theta}$ is jointly convex in $(\theta, b)$.

Let $(b_1, \theta_1)$ and $(b_2, \theta_2)$ any given pair of parameters and let $\lambda \in [0, 1]$ :

$$\int e^{(\lambda \theta_1 + (1-\lambda)\theta_2)^T s(x) - (\lambda b_1 + (1-\lambda)b_2)} \mathrm{d}x = \int e^{\lambda(\theta_1^T s(x) - b_1) + (1-\lambda)(\theta_2^T s(x) - b_2)} \mathrm{d}x \tag{59}$$

$$\geq \left[ \int \left( \lambda e^{\theta_1^T s(x) - b_1} + (1-\lambda)e^{\theta_2^T s(x) - b_2} \right) \mathrm{d}x \right] \tag{60}$$

$$= \lambda \int e^{\theta_1^T s(x) - b_1} \mathrm{d}x + (1-\lambda) \int e^{\theta_2^T s(x) - b_2} \mathrm{d}x. \tag{61}$$

The function $(\theta, b) \mapsto e^{-b + \log Z_\theta}$ is convex jointly in $(\theta, b)$, thus $(\theta, b) \mapsto \ell_{\mathrm{SNL}}(\theta, b)$ is also convex jointly in $(\theta, b)$ which concludes the proof. $\qquad \square$

## B  THE GAUSSIAN CASE

We consider a univariate Gaussian with unknown mean $\theta \in \mathbb{R}$ and known unit variance. The model is parameterised as an exponential family with energy and normalising constant:

$$E_\theta(x) = -\theta x, \ \log Z_\theta = \frac{1}{2}\theta^2, \tag{62}$$

the base measure being the standard Gaussian measure.

For a dataset $(x_1, ..., x_n) \in \mathbb{R}^n$, the log-likelihood is:

$$\ell(\theta) = \frac{1}{n} \sum_{i=1}^n x_i \theta - \frac{1}{2}\theta^2, \tag{63}$$

which is concave and is maximised at $\hat{\theta}_{\mathrm{ML}} = \bar{x}_n$. The SNL equals:

$$\ell_{\mathrm{SNL}}(\theta, b) = \frac{1}{n} \sum_{i=1}^n x_i \theta - b - Z_\theta e^{-b} + 1 \tag{64}$$

$$= \frac{1}{n} \sum_{i=1}^n x_i \theta - b - e^{\frac{1}{2}\theta^2 - b} + 1, \tag{65}$$

which is also concave and is maximised at $(\hat{\theta}_{\mathrm{SNL}}, \hat{b}_{\mathrm{SNL}}) = (\bar{x}_n, \bar{x}_n^2/2)$.

## C  THE BERNOULLI CASE

In the same vain as in Appendix B, we derive here the SNL for a Bernoulli distribution, in order to gain basic insights. We consider a Bernoulli distribtuion with unknown natural parameter $\theta \in \mathbb{R}$ ($\theta$ here is the logit of the probability of success). The model is parameterised as an exponential family with energy and normalising constant:

$$E_\theta(x) = -\theta x, \ \log Z_\theta = \log \left( 1 + e^\theta \right), \tag{66}$$

the base measure being the uniform measure on $\{0, 1\}$.

For a dataset $(x_1, ..., x_n) \in \{0, 1\}^n$, the log-likelihood is:

$$\ell(\theta) = \frac{1}{n} \sum_{i=1}^n x_i \theta - \log \left( 1 + e^\theta \right), \tag{67}$$

which is concave and is maximised at $\hat{\theta}_{\mathrm{ML}} = \mathrm{logit}(\bar{x}_n)$. The SNL equals:

$$\ell_{\mathrm{SNL}}(\theta, b) = \frac{1}{n} \sum_{i=1}^{n} x_i \theta - b - Z_\theta e^{-b} + 1 \tag{68}$$

$$= \frac{1}{n} \sum_{i=1}^{n} x_i \theta - b - e^{-b} \left(1 + e^\theta\right) + 1, \tag{69}$$

which is also concave and is maximised at $(\hat{\theta}_{\mathrm{SNL}}, \hat{b}_{\mathrm{SNL}}) = (\mathrm{logit}(\bar{x}_n), \log(1 + e^{\mathrm{logit}(\bar{x}_n)}))$.

## D  THE KULLBACK-LEIBLER DIVERGENCE FOR UN-NORMALISED DENSITIES

We consider a measured space $\mathcal{X}$, equipped with a base measure $\mathrm{d}x$ (typically the Lebesgue or the counting measure). Let $f_1, f_2$ be the densities of two finite measures. The Kullback-Leibler between these is then defined as

$$\mathrm{KL}(f_1 \| f_2) = \int \log \left(\frac{f_1(x)}{f_2(x)}\right) f_1(x) \mathrm{d}x + \left(\int f_2(x) \mathrm{d}x - \int f_1(x) \mathrm{d}x\right). \tag{70}$$

It is clear that this reduces to the usual KL when $f_1$ and $f_2$ are probability densities. For more details, see for instance Amari and Nagaoka (2000, Section 3.6) or Stummer and Vajda (2010).

Why is this a sensible generalisation? We can write our un-normalised densities as $f_1 = \mu_1 p_1$ and $f_2 = \mu_1 p_2$, where

$$\mu_1 = \int f_1(x) \mathrm{d}x, \ \mu_2 = \int f_2(x) \mathrm{d}x. \tag{71}$$

Plugging this into equation 70 gives

$$\mathrm{KL}(f_1 \| f_2) = \mu_1 \mathrm{KL}(p_1 \| p_2) + \mu_1 \log \left(\frac{\mu_1}{\mu_2}\right) + (\mu_2 - \mu_1) \tag{72}$$

$$= \mu_1 \left(\mathrm{KL}(p_1 \| p_2) + h \left(\frac{\mu_2}{\mu_1}\right)\right), \tag{73}$$

where $h : t \mapsto t - 1 - \log t$. Since $h(t) > 0$ for all $t \neq 1$ and $h(1) = 0$, we will have

- $\mathrm{KL}(f_1 \| f_2) \geq 0$
- $\mathrm{KL}(f_1 \| f_2) = 0$ if and only if $f_1 = f_2$.

This means that this generalised KL enjoys some of the nice properties of the usual KL, which motivates its use for statistical inference.

Another interesting property that is a direct consequence of Eq. (72) is that

$$\mathrm{KL}(p_1 \| p_2) = \min_{c > 0} \mathrm{KL}(p_1 \| c f_2), \tag{74}$$

which means that we can recover the KL between probability densities by minimising the KL between un-normalised densities, transforming the computation of the normalising constant into an optimisation problem. This justifies Eq. (19). Another interpretation of this property is that the KL between $p_1$ and the set $\{c f_2; c > 0\}$ is just the KL between $p_1$ and $p_2$.

The KL divergence between two un-normalised densities relates to the self-normalised log-likelihood as such:

$$\text{KL}(p_{\text{data}}||ce^{-E_\theta}) = \int \log\left(\frac{p_{\text{data}}(x)}{ce^{-E_\theta(x)}}\right) p_{\text{data}}(x)(x)\mathrm{d}x + cZ_\theta - 1 \tag{75}$$

$$= -\int\left(e^{-E_\theta(x)}p_{\text{data}}(x) - \log c + cZ_\theta - 1\right)\mathrm{d}x + \underbrace{\int \log(p_{\text{data}}(x))p_{\text{data}}(x)\mathrm{d}x}_{\text{does not depend on }\theta\text{ nor }c}. \tag{76}$$

As we assume we have access to an i.i.d. dataset $x_1, ..., x_n$, we can estimate the above quantity:

$$\text{KL}(p_{\text{data}}||ce^{-E_\theta}) \approx -\frac{1}{n}\sum_{i=1}^n e^{-E_\theta(x_i)} - \log c + cZ_\theta - 1 + \int \log(p_{\text{data}}(x))p_{\text{data}}(x)\,\mathrm{d}x \tag{77}$$

$$= -\ell_{\text{SNL}}(\theta, \log c) + \int \log(p_{\text{data}}(x))p_{\text{data}}(x)\,\mathrm{d}x. \tag{78}$$

This implies that maximising the self-normalised log-likelihood will, asymptotically, resemble minimising the generalised Kullback-Leibler divergence.

# E  ALGORITHMS

---

**Algorithm 1:**  Training an EBM for density estimation using SNL loss and proposal $q$.

---

**input**  : Learning iterations, $T$; learning rate, $\eta$; initial parameters, $\{\theta_0, b_0\}$; observed examples, $\{x_i\}_{i=1}^n$; batch size, $n_b$; number of samples from the proposal $q$, $M$.

**output** : $\theta_T, b_T$.

**for** $t = 0 : T - 1$ **do**

1. **Mini-batch**: Sample observed examples $\{x_i\}_{i=1}^{n_b}$.
2. **Proposal sampling**: Sample M elements from the proposal $x_m \sim \tilde{q}(x_m)$
3. **Learn EBM parameters** $\theta$: Update $\theta_{t+1} = \theta_t - \eta\hat{\nabla}_\theta\ell_{\text{SNL}}(\theta, b)$ using $\hat{\nabla}_\theta\ell_{\text{SNL}}(\theta_t, b)$ defined in Eq. (15).
4. **Learn** $b$: Update $b_{t+1} = b_t - \eta\hat{\nabla}_b\ell_{\text{SNL}}(\theta, b_t)$ using $\hat{\nabla}_b\ell_{\text{SNL}}(\theta, b_t)$ in defined Eq. (16).

---

**Algorithm 2:**  Training a VAE with EBM prior using the SNELBO loss.

---

**input**  : Learning iterations, $T$; learning rate, $\eta$; initial parameters, $\{\theta_0, \gamma_0, \phi_0, b_0\}$; observed examples, $\{x_i\}_{i=1}^n$; batch size, $n_b$; number of samples from the base $d$, $M$.

**output** : $\theta_T, \gamma_T, \phi_T, b_T$.

**for** $t = 0 : T - 1$ **do**

1. **Mini-batch**: Sample observed examples $\{x_i\}_{i=1}^{n_b}$.
2. **Proposal sampling**: Sample M elements from the base $x_m \sim \tilde{d}(x_m)$
3. **Learn EBM parameters** $\theta$: Update $\theta_{t+1} = \theta_t - \eta\hat{\nabla}_\theta\mathcal{L}_{\text{SNL}}((\theta, \gamma, \phi, b)$ using Eq. (26).
4. **Learn VAE parameters**: Update $\{\gamma, \phi\}_{t+1} = \{\gamma, \phi\}_t - \eta\hat{\nabla}_{\{\gamma,\phi\}}\mathcal{L}_{\text{SNL}}(\theta, \gamma, \phi, b)$ using Eq. (26).
5. **Learn** $b$: Update $b_{t+1} = b_t - \eta\hat{\nabla}_b\ell_{\text{SNL}}(\theta, b_t)$ using Eq. (26).

---

# F  DERIVATION OF THE SNELBO

Using the variational distribution $q_\gamma(z|x)$, we can write the regular ELBO for the VAE with the energy-based prior as

$$\mathcal{L}(\theta, \phi, \gamma) = \mathbb{E}_{q_\gamma(z|x)}[\log p_\phi(x|z)] + \mathbb{E}_{q_\gamma(z|x)}\left[\log\frac{e^{-E_\theta(z)}d(z)}{q_\gamma(z|x)Z_\theta}\right], \tag{79}$$

which is a lower bound, $\ell(\theta, \phi) \geq \mathcal{L}(\theta, \phi, \gamma)$, on the log-likelihood

$$\ell(\theta, \phi) = p_{\theta,\phi}(x) = \int p_\phi(x|z)p_\theta(z)\,\mathrm{d}z, \tag{80}$$

where we left out the sum over data to simply the notation. Using Lemma 2.1, we define the SNELBO as

$$
\mathcal{L}_{\text{SNL}}(\theta, \phi, \gamma, b) = \mathbb{E}_{q_\gamma(z|x)}[\log p_\phi(x|z)] + \mathbb{E}_{q_\gamma(z|x)}\left[\log \frac{d(z)}{q_\gamma(z|x)}\right]
$$
$$
+ \mathbb{E}_{q_\gamma(z|x)}\left[-E_\theta(z) - b\right] - Z_\theta e^{-b} + 1, \quad (81)
$$

which can be written using the base distribution $d$,

$$
\mathcal{L}_{\text{SNL}}(\theta, \phi, \gamma, b) = \mathbb{E}_{q_\gamma(z|x)}\left[\log p_\phi(x|z)\right] + \mathbb{E}_{q_\gamma(z|x)}\left[\log \frac{d(z)}{q_\gamma(z|x)}\right]
$$
$$
+ \mathbb{E}_{q_\gamma(z|x)}\left[-E_\theta(z) - b\right] - \mathbb{E}_{d(z)}\left[e^{-E_\theta(z)-b}\right] + 1 \quad (82)
$$

Lemma 2.1 gives directly the following results :

$$
\ell(\theta, \phi) \geq \mathcal{L}(\theta, \phi, \gamma) \geq \mathcal{L}_{\text{SNL}}(\theta, \phi, \gamma, b) \tag{83}
$$

## G  REGRESSION DATASETS

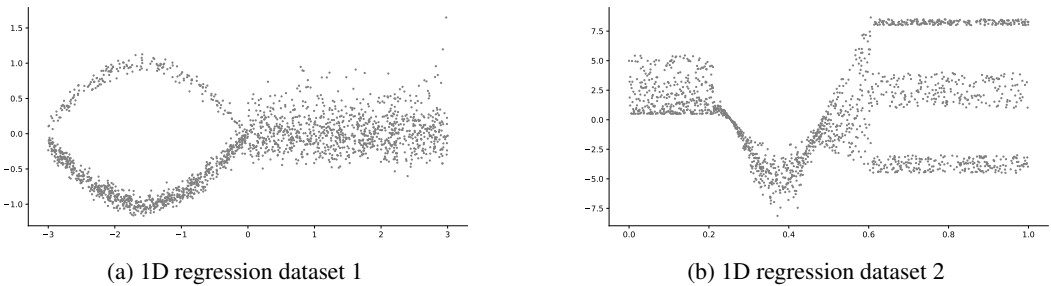

(a) 1D regression dataset 1          (b) 1D regression dataset 2

Figure 3: Visualisation of the two toy regression datasets.

The first dataset set, on the left hand side of Fig. 3, is a mixture of two gaussians with weights $0.2$ and $0.8$ for negative values on the $x$-axis and a log-normal distribution $\text{Log} -\mathcal{N}(0., 0.25)$ for positive values of $x$. There are 2000 training samples that we generated by uniformly sampling values in $[-3, 3]$.

The second dataset, on the right hand side of Fig. 3, is defined for $x$ in $[0, 1]$ and is divided in four different chunks. The first one, for $x < 0.21$, is sampled from $\text{Beta}(\alpha = 0.5, \beta = 1)$; the second one, for $0.21 \leq x < 0.47$ is sampled from $\mathcal{N}\left(\mu = 3 \cdot \cos x - 2, \sigma = |3 \cdot \cos x - 2|\right)$; the third one for $0.47 \leq x < 0.61$ from an increasing uniform distribution; the fourth and last one, for $0.61 \leq x \leq 1$ is obtained from a mixture of uniform distribution, $\mathcal{U}(8, 0.5), \mathcal{U}(1, 3)$ and $\mathcal{U}(-4.5, 1.5)$.

## H  ADDITIONAL RESULTS

| Models | | | | Datasets | | | |
| --- | --- | --- | --- | --- | --- | --- | --- |
| | | | | Regression Dataset 1 | | Regression Dataset 2 | |
| Objective | Proposal $q$ | $b_\phi$ | Base Dist | $\ell_{\text{IS}}$ | $\ell_{\text{SNL}}$ | $\ell_{\text{IS}}$ | $\ell_{\text{SNL}}$ |
| NCE | $\mathcal{N}(\mu,\Sigma)$ | None | None | $-0.100\,_{(\pm 0.186)}$ | $-0.638\,_{(\pm 0.168)}$ | $-2.416\,_{(\pm 0.376)}$ | $-3.049\,_{(\pm 0.900)}$ |
| NCE | $\mathcal{N}(\mu,\Sigma)$ | None | $q$ | $-0.336\,_{(\pm 0.468)}$ | $-1.567\,_{(\pm 0.282)}$ | $-2.548\,_{(\pm 0.232)}$ | $-2.676\,_{(\pm 0.169)}$ |
| NCE | $\mathcal{N}(\mu,\Sigma)$ | MLP | None | $-0.030\,_{(\pm 0.278)}$ | $-0.718\,_{(\pm 0.256)}$ | $-2.592\,_{(\pm 0.214)}$ | $-3.559\,_{(\pm 1.881)}$ |
| NCE | $\mathcal{N}(\mu,\Sigma)$ | MLP | $q$ | $-0.644\,_{(\pm 0.632)}$ | $-1.580\,_{(\pm 0.480)}$ | $-2.426\,_{(\pm 0.257)}$ | $-2.586\,_{(\pm 0.238)}$ |
| NCE | MDN K2 | None | None | $-0.570\,_{(\pm 0.209)}$ | $-1.275\,_{(\pm 0.688)}$ | $-2.451\,_{(\pm 0.040)}$ | $-3.094\,_{(\pm 0.515)}$ |
| NCE | MDN K2 | MLP | None | $-0.611\,_{(\pm 0.154)}$ | $-1.492\,_{(\pm 0.993)}$ | $-2.451\,_{(\pm 0.088)}$ | $-2.634\,_{(\pm 0.084)}$ |
| SNL | $\mathcal{N}(\mu,\Sigma)$ | None | None | $0.091\,_{(\pm 0.122)}$ | $-0.023\,_{(\pm 0.071)}$ | $-1.597\,_{(\pm 0.047)}$ | $-1.619\,_{(\pm 0.063)}$ |
| SNL | $\mathcal{N}(\mu,\Sigma)$ | None | $q$ | $0.065\,_{(\pm 0.084)}$ | $-0.044\,_{(\pm 0.095)}$ | $-1.493\,_{(\pm 0.039)}$ | $-1.503\,_{(\pm 0.041)}$ |
| SNL | $\mathcal{N}(\mu,\Sigma)$ | MLP | None | $0.164\,_{(\pm 0.088)}$ | $0.033\,_{(\pm 0.077)}$ | $-1.813\,_{(\pm 0.109)}$ | $-1.836\,_{(\pm 0.109)}$ |
| SNL | $\mathcal{N}(\mu,\Sigma)$ | MLP | $q$ | $0.091\,_{(\pm 0.094)}$ | $-0.048\,_{(\pm 0.030)}$ | $\mathbf{-1.468}\,_{(\pm 0.014)}$ | $\mathbf{-1.477}\,_{(\pm 0.016)}$ |
| SNL | MDN K2 | None | None | $0.227\,_{(\pm 0.058)}$ | $0.221\,_{(\pm 0.059)}$ | $-2.061\,_{(\pm 0.145)}$ | $-2.070\,_{(\pm 0.141)}$ |
| SNL | MDN K2 | MLP | None | $\mathbf{0.255}\,_{(\pm 0.017)}$ | $\mathbf{0.251}\,_{(\pm 0.016)}$ | $-2.099\,_{(\pm 0.250)}$ | $-2.170\,_{(\pm 0.353)}$ |

Table 6: Evaluation of regression EBMs on the 1D toy regression problems with two different objectives and different sets of parameters. Each model is trained for five runs and we report the mean and standard deviation of the estimated log-likelihood $\ell_{\text{IS}}$ and the self normalized log-likelihood $\ell_{\text{SNL}}$. Using the SNL as objective clearly outperforms the NCE.

| Models | | Datasets | | | | | | | |
| --- | --- | --- | --- | --- | --- | --- | --- | --- | --- |
| | | Steering Angle | | Cell Count | | UTKFaces | | BIWI | |
| Objective | Proposal | $\ell_{\text{IS}}$ | $\ell_{\text{SNL}}$ | $\ell_{\text{IS}}$ | $\ell_{\text{SNL}}$ | $\ell_{\text{IS}}$ | $\ell_{\text{SNL}}$ | $\ell_{\text{IS}}$ | $\ell_{\text{SNL}}$ |
| NCE | $\mathcal{N}(\mu,\Sigma)$ | $-3.649\,_{(\pm 1.224)}$ | UNNORMALIZED | $-3.367\,_{(\pm 0.399)}$ | $-9.675\,_{(\pm 0.605)}$ | $-3.147\,_{(\pm 0.1100)}$ | $-8.223\,_{(\pm 3.795)}$ | $-11.02\,_{(\pm 0.576)}$ | UNNORMALIZED |
| NCE | MDN-4 | $-4.044\,_{(\pm 0.741)}$ | $-10.272\,_{(\pm 0.742)}$ | $-3.856\,_{(\pm 0.029)}$ | UNNORMALIZED | $-3.876\,_{(\pm 0.140)}$ | $-4.821\,_{(\pm 0.233)}$ | $-12.093\,_{(\pm 0.155)}$ | UNNORMALIZED |
| NCE | MDN-8 | $-4.001\,_{(\pm 0.667)}$ | UNNORMALIZED | $-3.864\,_{(\pm 0.048)}$ | UNNORMALIZED | $-4.123\,_{(\pm 0.21)}$ | $-5.170\,_{(\pm 0.955)}$ | $-11.998\,_{(\pm 0.339)}$ | UNNORMALIZED |
| SNL | $\mathcal{N}(\mu,\Sigma)$ | $-2.665\,_{(\pm 1.37)}$ | $-3.973\,_{(\pm 3.15)}$ | $-2.701\,_{(\pm 0.041)}$ | $-2.725\,_{(\pm 0.046)}$ | $-2.966\,_{(\pm 0.057)}$ | $-2.991\,_{(\pm 0.069)}$ | $-10.86\,_{(\pm 1.017)}$ | $-11.05\,_{(\pm 1.141)}$ |
| SNL | Uniform | $\mathbf{-1.402}\,_{(\pm 0.068)}$ | $\mathbf{-1.423}\,_{(\pm 0.074)}$ | $\mathbf{-2.604}\,_{(\pm 0.001)}$ | $\mathbf{-2.620}\,_{(\pm 0.007)}$ | $-2.927\,_{(\pm 0.032)}$ | $-2.965\,_{(\pm 0.019)}$ | $-10.44\,_{(\pm 0.138)}$ | $-10.51\,_{(\pm 1.222)}$ |
| SNL | MDN-4 | $-1.780\,_{(\pm 0.2312)}$ | $-1.795\,_{(\pm 0.231)}$ | $-2.834\,_{(\pm 0.041)}$ | $-2.846\,_{(\pm 0.043)}$ | $-2.992\,_{(\pm 0.045)}$ | $-3.004\,_{(\pm 0.075)}$ | $-10.08\,_{(\pm 0.149)}$ | $-10.11\,_{(\pm 0.126)}$ |
| SNL | MDN-8 | $-1.673\,_{(\pm 0.042)}$ | $-1.692\,_{(\pm 0.046)}$ | $-2.801\,_{(\pm 0.071)}$ | $-2.811\,_{(\pm 0.071)}$ | $\mathbf{-2.921}\,_{(\pm 0.055)}$ | $\mathbf{-2.943}\,_{(\pm 0.062)}$ | $\mathbf{-10.01}\,_{(\pm 0.092)}$ | $\mathbf{-10.04}\,_{(\pm 0.091)}$ |

Table 7: Evaluation of EBMs for regression on image regression datasets with two different objectives and different proposals. Each model is trained for five runs and we report the mean and standard deviation of the estimated log-likelihood ($\ell_{IS}$) and estimated self-normalised log-likelihood ($\ell_{SNL}$). When the proposal is MDN, the proposal is learned jointly with the model following Gustafsson et al. (2022).

# I DESCRIPTION OF THE NEURAL NETWORKS

## I.1 2D DISTRIBUTION ESTIMATION

| $E_\theta$ | Activation | Output shape |
|---|---|---|
| Fully Connected | ReLU | $2 \times 200$ |
| Fully Connected | ReLU | $200 \times 100$ |
| Fully Connected | ReLU | $100 \times 50$ |
| Fully Connected | ReLU | $50 \times 50$ |
| Fully Connected | ReLU | $50 \times 1$ |
| Total trainable parameters | | 30450 |

Table 8: $E_\theta$ for the toy distribution estimation

## I.2 1D REGRESSION

| Feature extractor | Activation | Output shape |
|---|---|---|
| Fully Connected | ReLU | $10 \times 10$ |
| Fully Connected | ReLU | $10 \times 10$ |
| Fully Connected | ReLU | $10 \times 1$ |
| Total trainable parameters | | 210 |

Table 9: Feature extractor. Inputs $x$ and outputs $h_x$

| $E_\theta$ | Activation | Output shape |
|---|---|---|
| Input $y \to$ Output $f(y)$ | | |
| Fully Connected | ReLU | $1 \times 16$ |
| Fully Connected | ReLU | $16 \times 32$ |
| Fully Connected | ReLU | $32 \times 64$ |
| Fully Connected | ReLU | $64 \times 128$ |
| Concatenation of $h_x$ and $f(y)$ | | |
| Fully Connected | ReLU | $144 \times 10$ |
| Fully Connected | ReLU | $10 \times 1$ |
| Total trainable parameters | | 30450 |

Table 10: $E_\theta$ for 1d regression estimation.

| MDN | Activation | Output shape |
|---|---|---|
| Input $h_x$ | | |
| Fully Connected | ReLU | $10 \times 10$ |
| Fully Connected | ReLU | $10 \times K$ |
| Total trainable parameters | | $100 + 10 \times K$ |

Table 11: Neural network estimating one parameter of the MDN with $K$ components in the mixture.

| $b_\phi$ | Activation | Output shape |
|---|---|---|
| Input $h_x$ | | |
| Fully Connected | ReLU | $10 \times 10$ |
| Fully Connected | ReLU | $10 \times 1$ |
| Total trainable parameters | | 110 |

Table 12: Neural network estimating the normalization constant $Z_{\theta,x}$ for every $x$.

### I.3  IMAGE REGRESSION

The feature extractor is a Resnet-18 He et al. (2016) from torchvision Paszke et al. (2019).

| $E_\theta$ | Activation | Output shape |
|---|---|---|
| Input $y \to$ Output $f(y)$ | | |
| Fully Connected | ReLU | $1 \times 16$ |
| Fully Connected | ReLU | $16 \times 32$ |
| Fully Connected | ReLU | $32 \times 64$ |
| Fully Connected | ReLU | $64 \times 128$ |
| Concatenation of $h_x$ and $f(y)$ | | |
| Fully Connected | ReLU | $640 \times 640$ |
| Fully Connected | ReLU | $640 \times 1$ |
| Total trainable parameters | | 420816 |

Table 13: $E_\theta$ for 1d regression estimation.

| MDN | Activation | Output shape |
|---|---|---|
| Input $h_x$ | | |
| Fully Connected | ReLU | $512 \times 512$ |
| Fully Connected | ReLU | $512 \times K$ |
| Total trainable parameters | | $262144 + 512 \times K$ |

Table 14: Neural network estimating one parameter of the MDN with $K$ components in the mixture. We use three such networks for $\pi_\psi, \mu_\psi, \sigma_\psi$.

| $b_\phi$ | Activation | Output shape |
|---|---|---|
| Input $h_x$ | | |
| Fully Connected | ReLU | $512 \times 512$ |
| Fully Connected | ReLU | $512 \times 1$ |
| Total trainable parameters | | 262656 |

Table 15: Neural network estimating the normalization constant $Z_{\theta,x}$ for every $x$.

### I.4  VAE WITH PRIOR EBM

| EBM Model for BinaryMNIST | | |
|---|---|---|
| Layers | In-Out Size | Stride |
| Input: $z$ | 100 | |
| Linear, LReLU | 200 | - |
| Linear, LReLU | 200 | - |
| Linear | 1 | - |
| **Generator Model for BinaryMNIST**, ngf = 16 | | |
| Input: $z$ | 16×1×1 | |
| 4×4 convT(ngf × 8), LReLU | 4×4×(ngf × 8) | 1 |
| 3×3 convT(ngf × 4), LReLU | 7×7×(ngf × 4) | 2 |
| 4×4 convT(ngf × 2), LReLU | 14×14×(ngf × 2) | 2 |
| 4×4 convT(3), Sigmoid | 28×28×1 | 2 |
| **Encoder Model for BinaryMNIST**, ngf = 16 | | |
| Input: × | 1×28×28 | |
| 5×5 conv(ngf × 2), LReLU | 14×14×(ngf × 2) | 2 |
| 5×5 conv(ngf × 4), LReLU | 7×7×(ngf × 4) | 2 |
| 5×5 conv(ngf × 8), LReLU | 4×4×(ngf × 8) | 2 |
| Linear, - | 16 | - |

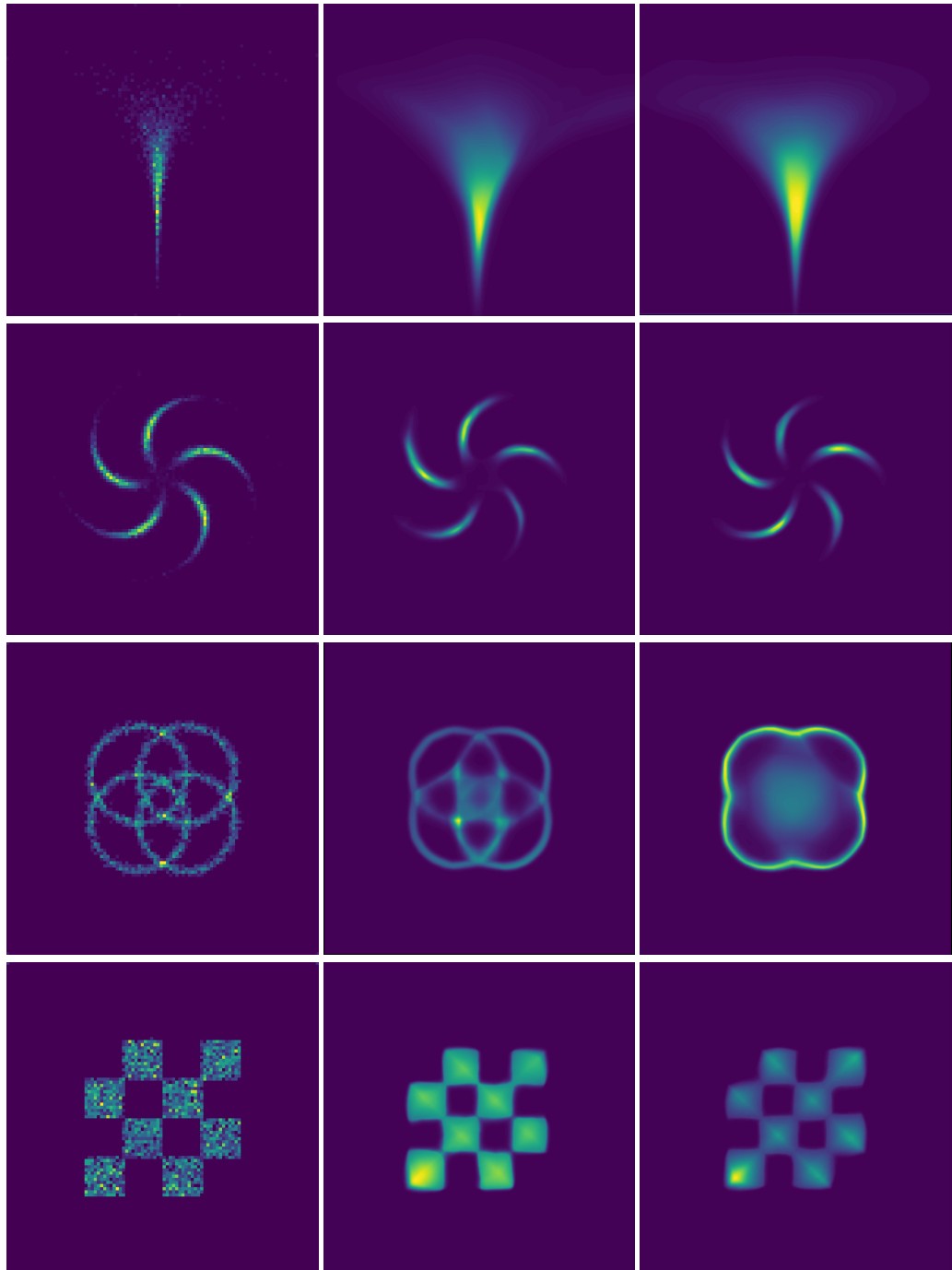

Figure 4: Each row is a dataset, the first column displays samples from the dataset, the second column displays the energy function of an EBM trained with the self normalised log-likelihood (ours), the third column displays the energy function of an EBM trained with NCE. We use a standard Gaussian as base distribution for both training methods. These parameterisations corresponds to the first two lines of Table 1.

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
