# OpenReview forum: "Learning energy-based models by self-normalising the likelihood"
_ICLR.cc/2024/Conference — Submitted to ICLR 2024_

### Official Review · Reviewer_qYkm · 2023-10-21

**Soundness:** 3 good
**Presentation:** 2 fair
**Contribution:** 3 good
**Rating:** 5
**Confidence:** 3

**Summary:**

This paper studies the learning of the energy-based model (EBM). The typical MLE learning requires MCMC sampling of EBM to obtain the gradient of the normalizing function, which can be challenging in practice due to its instability and computational cost. The proposed self-normalized log-likelihood (SNL) method instead parameterizes the normalizing function via a linear variation formulation (Eq.8 in the paper), which does not involve the MCMC sampling but can train the EBM with the SNL estimate.

**Strengths:**

1. The energy-based model serves as a foundational generative model, and the proposed learning algorithm is thus well-motivated.
2. The paper is in general well-presented, especially the theoretical parts regarding the understanding of the proposed method.
3. The proposed method seems to be flexible as the author extends it to multiple settings, such as prior of VAE and regression tasks (in a supervised scenario).

**Weaknesses:**

1. This paper has a well-motivated idea and contains comprehensive theoretical derivation for understanding the key idea. However, as mentioned by the author, the NCE method is related, it would be nice to have a deeper theoretical connection and comparison with the NCE method. For now, the major comparison is shown by empirical experiments.
2. Many other prior works can be applied to some more challenging real data, such as CIFAR-10, CelebA-64, or even the high-resolution (CelebA-HQ-256), so what limited this learning algorithm for such dataset?
3. As a novel learning method, it would be nice to have a practical learning algorithm to simplify and illustrate the main idea.

**Questions:**

(1) How do we understand the unbiased estimate of SNL (Eq.15) while the last term is based on Jensen's (Eq.6)?

(2) Some typos can be fixed (e.g., Eq.17 \nabla_\theta missing)

---

> ### Author Response · Authors · 2023-11-19
>
> We want to thank reviewer qYkm for their careful analysis of our paper and suggestions for improvement.
>
>
> *1 - This paper has a well-motivated idea and contains comprehensive theoretical derivation for understanding the key idea. However, as mentioned by the author, the NCE method is related, it would be nice to have a deeper theoretical connection and comparison with the NCE method. For now, the major comparison is shown by empirical experiments.*
>
>
> Thank you for suggesting an investigation of the relationship between the NCE and SNL objectives. We found that the SNL loss resembles one of the generalisations of NCE proposed in a theoretical paper on NCE [2]. We will discuss this paper in the revised version.
>
>
>
> *2 - Many other prior works can be applied to some more challenging real data, such as CIFAR-10, CelebA-64, or even the high-resolution (CelebA-HQ-256), so what limited this learning algorithm for such dataset?*
>
> The goal of our paper is to introduce a new method for training Energy-Based Models. As it was done for other methods in the field, scaling such methods requires tuning and tricks that are a completely new avenue of research. For image modelling, we will require more complicated tuning and proposal than just a simple Gaussian maybe using a flow as a proposal. For instance [1] focuses on MCMC-based methods but scales it using a replay buffer.
>
>
>
> *3 - As a novel learning method, it would be nice to have a practical learning algorithm to simplify and illustrate the main idea.*
>
> We added an algorithm description for our method in appendix E. Thank you for the suggestions.
>
>
> [1] Du, Y. and Mordatch, I. "Implicit Generation and Modeling with Energy Based Models "Advances in Neural Information Processing Systems 32 (2019)
>
> [2] Pihlaja et al. (2010), A family of computationally efficient and simple estimators for unnormalized statistical models, Uncertainty in Artificial Intelligence

---

> > ### Comment · Reviewer_qYkm · 2023-11-22
> >
> > I would like to thank the authors for their responses, but it seems that my concern is not well addressed in detail, such as the connection to the NCE method.
> >
> > I understand real image datasets can be challenging, but I honestly don't think this is a "complete" new avenue of research. [1] propose a "replay buffer" as an improvement for the performance and many other works (e.g., [a] using only short-run Langevin dynamics) still work without the replay buffer. Therefore, I do think applying the proposed method to some standard benchmark (e.g., CIFAR-10 32x32) can be better for illustrating the potential of the method and motivating future research of such an active field.
> >
> >
> >
> >
> > [a] Nijkamp, Erik, et al. "Learning non-convergent non-persistent short-run MCMC toward energy-based model." Advances in Neural Information Processing Systems 32 (2019)

---

### Official Review · Reviewer_3VY8 · 2023-10-28

**Soundness:** 4 excellent
**Presentation:** 4 excellent
**Contribution:** 3 good
**Rating:** 5
**Confidence:** 4

**Summary:**

In this paper, the authors introduce a novel objective function for estimating Energy-Based Models (EBM), offering a promising alternative to the costly Markov Chain Monte Carlo (MCMC) sampling method. Their proposed objective function not only eliminates the need for MCMC but also provides an estimate of the log-normalizing constant as a byproduct. The paper showcases the effectiveness of this new approach by demonstrating its ability to recover the Maximum Likelihood (ML) estimate and its favorable performance within the exponential family of distributions. To validate their method, the authors conduct comprehensive empirical tests on both low-dimensional synthetic and real-world datasets, illustrating its efficacy and its superiority over Noise Contrastive Estimation (NCE). Additionally, the authors extend the application of their method to Variational Autoencoders (VAEs) with energy-based priors, broadening the scope of their contribution in the field of generative models.

**Strengths:**

- The paper is well written
- The idea of doing a variational approximation of the logarithm is elegant
- The application to VAEs with energy-based priors is interesting

**Weaknesses:**

- The method seems very sensitive to the curse of dimensionality because of its IS component. This scaling issue is not investigated.
- The proposed method is not compared against MCMC-based methods.
- The sensitivity to the choice of proposal should be critical but it is only investigated in low-dimensional cases.
- Most experiments are toy experiments or in a very low dimension.

**Questions:**

1. Can you provide more real-world experiments ? For instance in generative modeling (without the VAE component) or out-of-distribution detection.

2. As mentioned in the weaknesses, I would expect your method to be very sensitive to the design of $q$.

(a) Did you run a sensitivity comparison with [1], [2] (which develop similar ideas) or NCE in higher dimensional settings (compared to Sec 4.2) ?

(b) Did you try to learn $q$ as done in [3] for NCE ?

3. In [4], the authors give a very similar result as your theorem 3.1 but for NCE. Is there more theoretical comparisons to be drawn against NCE ?

4. As mentioned in the weaknesses, I think it would be nice to compare SNL against MCMC-based methods (at least Langevin based) with apple-to-apple computational budgets.

[1] Will Grathwohl, Jacob Kelly, Milad Hashemi, Mohammad Norouzi, Kevin Swersky, & David Duvenaud. (2021). No MCMC for me: Amortized sampling for fast and stable training of energy-based models.

[2] Hanjun Dai, Rishabh Singh, Bo Dai, Charles Sutton, & Dale Schuurmans. (2020). Learning Discrete Energy-based Models via Auxiliary-variable Local Exploration.

[3] Ruiqi Gao, Erik Nĳkamp, Diederik P. Kingma, Zhen Xu, Andrew M. Dai, & Ying Nian Wu. (2020). Flow Contrastive Estimation of Energy-Based Models.

[4] Bingbin Liu, Elan Rosenfeld, Pradeep Ravikumar, & Andrej Risteski. (2021). Analyzing and Improving the Optimization Landscape of Noise-Contrastive Estimation.

---

> ### Author Response · Authors · 2023-11-19
>
> We thank reviewer 3VY8 for their study of our paper and the recommendation for future iterations of the paper.
>
> *1 - Can you provide more real-world experiments? For instance in generative modeling (without the VAE component) or out-of-distribution detection.*
> *2 - As mentioned in the weaknesses, I would expect your method to be very sensitive to the design of $q$.
> (a) Did you run a sensitivity comparison with [1], [2] (which develop similar ideas) or NCE in higher dimensional settings (compared to Sec 4.2)?
> (b) Did you try to learn as done in [3] for NCE ?*
>
> Answer for *1* and *2*: We agree that the method will be sensitive to the proposal, especially when scaling it for image modelling. However, we feel that the work required for finding and tuning a correct proposal is a new avenue of work in itself. Indeed, all these papers are mostly focused on scaling existing methods by finding and tuning a proposal (noisy neural proposal in [1] using $l_{IS}$ in our paper or flow proposal in [3] for NCE). Note that we did compare $l_{IS}$ and $l_{SNL}$ with a Gaussian for the 2D toy experiment and the image regression experiments. In such cases, training the EBM was less stable than with our method or NCE.
>
>
>
> *3 - In [4], the authors give a very similar result as your theorem 3.1 but for NCE. Are there more theoretical comparisons to be drawn against NCE ?*
>
>
> The paper you are mentioning also provides more theoretical insights into the loss landscape for NCE. Conducting such a study for SNL would require substantial work but is an interesting new avenue of work but we will integrate it to a revised version of the paper. Thank you for suggesting an investigation of the relationship between the NCE and SNL objectives. We do not know if there is a direct relationship between our loss and the NCE one. However, we found that the SNL loss resembles one of the generalisations of NCE proposed in a theoretical paper on NCE [5]. We will discuss this paper in the revised version.
>
> *4 - As mentioned in the weaknesses, I think it would be nice to compare SNL against MCMC-based methods (at least Langevin based) with apple-to-apple computational budgets.*
>
> Due to the lack of time required to implement and tune MCMC-based methods, we will provide a comparison of SNL with MCMC based method using similar computational budgets (ie number of call of a neural networks and time) upon acceptance or for a further iteration of the paper.
>
> [1] Will Grathwohl, Jacob Kelly, Milad Hashemi, Mohammad Norouzi, Kevin Swersky, & David Duvenaud. (2021). No MCMC for me: Amortized sampling for fast and stable training of energy-based models.
>
> [2] Hanjun Dai, Rishabh Singh, Bo Dai, Charles Sutton, & Dale Schuurmans. (2020). Learning Discrete Energy-based Models via Auxiliary-variable Local Exploration.
>
> [3] Ruiqi Gao, Erik Nĳkamp, Diederik P. Kingma, Zhen Xu, Andrew M. Dai, & Ying Nian Wu. (2020). Flow Contrastive Estimation of Energy-Based Models.
>
> [4] Bingbin Liu, Elan Rosenfeld, Pradeep Ravikumar, & Andrej Risteski. (2021). Analyzing and Improving the Optimization Landscape of Noise-Contrastive Estimation.
>
> [5] Pihlaja et al. (2010), A family of computationally efficient and simple estimators for unnormalized statistical models, Uncertainty in Artificial Intelligence

---

### Official Review · Reviewer_CuDM · 2023-11-01

**Soundness:** 2 fair
**Presentation:** 3 good
**Contribution:** 2 fair
**Rating:** 5
**Confidence:** 4

**Summary:**

This paper proposes to change the loss function of energy based model based on a variational formulation. The proposed algorithm estimate an extra parameter b together with the energy function $E_\theta(x)$. The author use importance sampling with base distribution $q(x)$ to estimate the normalizing term of the energy function. They carry out the experiments on toy example datasets and energy regression task.

**Strengths:**

I think this paper is well written and the idea is easy to follow. The reformulation trick, though simple, is interesting to me.

**Weaknesses:**

However, I am not fully convinced by whether the proposed algorithm really works better in practice than original EBM training when modeling complex distributions. This lies in several aspects:

1. From my own experience, the most challenging part when training the EBM is to get valid samples from the current fitted distribution to estimate the (gradient of) normalizing constant. Previous works try to solve this problem with different sampling techniques. While this work proposes a linear lower bound, it still needs to estimate the normalizing constant with Monte Carlo based method. Thus, it might not really alleviate the training difficulties.

2. To do this Monte Carlo estimation, the work employs important sampling using a base distribution $q(x)$ and $q(x)$ are simple distributions like Gaussian. I suspect this algorithm works because the target distributions tested in this work are very simple, either toy distribution or conditional distribution $p_\theta(y|X)$ where y is low dimensional. If we are modeling model complex distribution like unconditional distribution $p(x)$ on high dimensional data like images, then we still need Monte Carlo based methods and the previous diffculties are still there.

3. The proposed algorithm introduces a variational parameter b, and it requires to update b together with the energy function iteratively. Then similar to the VAE case, whether there can be a mismatch between the estimate of b and the energy function $E_\theta(X)$.  (Not sure whether the $\exp^{-b}$ term will make the training more unstable if b is not well optimized.) Or in other words, how diffcult is it to design the schedule of updating b and energy function to make this algorithm work.

4. As also mentioned in 2, the modeled distributions in the experiments are too simple to be convincing to me. The modeled experiments are either unconditional distribution on toy data or with image input but only models the conditional distribution on some low dimensional label. The VAE experiment in 5.3 models binary MNIST (which is also not very complex). And with the help of encoder and decoder, the latent space might be more simple. (Beside, what if we train the model VAE-EBM not with $l_{snl}$ but with plain MLE loss? There seems to be included as a baseline in Table 5.) I think in order to make the proposed algorithm more convincing, the authors need to demonstrate better results than pure MLE loss on more complex distributions like real image (face or cifar or SVHN).

5. The review for EBM study seems to be insuffcient, may consider the following works:

[1] Improved contrastive divergence training of energy-based models.

[2] Learning energy-based models by diffusion recovery likelihood.

[3] A tale of two flows: Cooperative learning of langevin flow and normalizing flow toward energy-based model.

**Questions:**

Please see weakness.

---

> ### Author Response · Authors · 2023-11-19
>
> We thank the reviewer CuDM for their careful review of our paper and address their concern below.
>
> *1 - From my own experience, the most challenging part when training the EBM is to get valid samples from the current fitted distribution to estimate the (gradient of) normalizing constant. [...]*
>
> Our experience matches the one of reviewer CuDM, where the difficulty is to get good samples from the current fitted distribution. In MCMC-based methods training the EBM requires long chains or tricks (for instance, keeping a buffer in [1]) to avoid biased training. However, our method does not require sampling from the model to estimate gradients and that is why we believe this method is more flexible and avoids biases training loss.
>
> *2 - To do this Monte Carlo estimation, the work employs important sampling using a base distribution and simple distributions like Gaussian. I suspect this algorithm works because the target distributions tested in this work are very simple,[...] .*
>
> Though we agree that having a simple distribution like a Gaussian might not scale to more complicated high-dimensional data like images, it does not necessarily mean the need to resort to MCMC-based methods again. Indeed, having a more complex proposal distribution like a flow as a proposal or base distribution could be a way to handle this issue.
>
>
>  *3 - The proposed algorithm introduces a variational parameter b, which requires updating b and the energy function iteratively. Then similar to the VAE case, whether there can be a mismatch between the estimate of b and the energy function. (Not sure whether the term will make the training more unstable if b is not well optimized.) Or in other words, how difficult is it to design the schedule of updating b and energy function to make this algorithm work?*
>
> It is true that the quality of the gradient estimate depends on how close $b$ is to the normalization constant. This is clearly seen in equation (17) of the paper where we rewrite SNL gradients as the likelihood gradients with some negligible terms if $b$ verifies the aforementioned condition :
> $$\begin{aligned}
>     \nabla_{\theta} \ell_{\text{snl}}(\theta,b) &=  - \frac{1}{n} \sum_{i=1}^{n}  \nabla_{\theta} E_{\theta}(x) - e^{-b+\log{Z_{\theta}}} \nabla \log{Z_{\theta}} \\
>     & = \nabla_{\theta} \ell_{\theta} + \nabla_{\theta} \log{Z_{\theta}} (1-e^{-b+\log{Z_{\theta}}}).
> \end{aligned}
> $$
> In practice, we train both $\theta$ and $b$ within the same iteration and very quickly during training, $b$ gets very close to $\log{Z_{\theta}}$ thus this did not require more tuning from our side. Note that we treat the extra parameter $b$ as the bias of the last layer of neural network and as such as any other parameter weight of the neural network, optimized using Adam.
>
>
>
> *4 - As also mentioned in 2, the modelled distributions in the experiments are too simple to be convincing to me. The modelled experiments are either unconditional distribution on toy data or with image input but only model the conditional distribution on some low dimensional label. The VAE experiment in 5.3 models binary MNIST (which is also not very complex).[...]*
>
> Though we agree that the experiments are low-dimensional, we disagree that they are not convincing enough. Indeed, to the best of our knowledge, this is the first time an EBM is used for density estimation (and actually provides an upper and lower bound of the likelihood) on the UCI datasets. These datasets usually require complicated networks (autoregressive flows for instance), whereas we reach similar results with much simpler networks and a simple Gaussian proposal. Similarly, we show state-of-the-art performance on EBMs for image regression which is in itself useful [3] for visual tracking or pose-estimation in computer vision.
> Finally, the point of the paper is to introduce a new method for training general EBMs, without a specific focus on high-dimensional data. As it was done for other methods in the field, scaling such methods requires tuning and tricks that are a completely new avenue of research.
>
> *4 bis - I think in order to make the proposed algorithm more convincing, the authors need to demonstrate better results than pure MLE loss on more complex distributions like real images (face or cifar or SVHN).*
>
> Could you specify what you call pure MLE loss? Is that just the standard VAE?
>
>
>
>
> *5-* We added the recommended paper to the references, we want to thank the reviewer for their valuable suggestions.
>
>
>
>
> [1] Du, Y. and Mordatch, I. "Implicit Generation and Modeling with Energy Based Models "Advances in Neural Information Processing Systems 32 (2019)
>
> [2] Will Grathwohl, et al. "No MCMC for me: Amortized samplers for fast and stable training of energy-based models." (2021).
>
> [3]  Gustafsson, F. K., Danelljan, M., Timofte, R., & Schön, T. B. (2020). How to train your energy-based model for regression. arXiv preprint arXiv:2005.01698.

---

> > ### Comment · Reviewer_CuDM · 2023-11-20
> >
> > I'd like to thank the authors for their responses. I want to make a clarification for my following question:
> >
> > #### "Beside, what if we train the model VAE-EBM not with  $l_{snl}$ but with plain MLE loss? I think in order to make the proposed algorithm more convincing, the authors need to demonstrate better results than pure MLE loss on more complex distributions like real image (face or cifar or SVHN)."
> >
> > In the current paper, Table 5 presents four scenarios: two without EBM, one with EBM trained post-hoc following the VAE training (VAE-EBM Post-Hoc), and one where EBM is trained concurrently with the VAE model (VAE-EBM) I believe when training the EBM, the authors use the $l_{snl}$ loss as described in their paper. I'm curious what if we train EBM together with VAE, but we do not use $l_{snl}$ loss. Instead, we use the plain maximum likelihood loss as shown in eq 3 and 4. (As done in [1].) From current results, I can see that the results with EBMs are better than those without EBM. And training EBM together with VAE works better than Post-Hoc training. However, this doesn't necessarily imply that the EBM trained with the techniques proposed in this paper ($l_{snl}$ loss) is superior to one trained with the standard maximum likelihood loss, as practiced in previous studies[1]. Therefore, I'd like to explore this question further. Also, I further found that the performance is measured by SNELBO. Will this metric favor the model trained with $l_{snl}$? What if pure ELBO is reported?
> >
> > [1] Learning latent space energy-based prior model.

---

> ### Author Response · Authors · 2023-11-22
>
> Thank you for clarifying your question.
>
> > Also, I further found that the performance is measured by SNELBO. Will this metric favor the model trained with ? What if pure ELBO is reported?
>
> Thank you for your question. "Pure" ELBO is actually reported when it is possible to report it (ie for Vanilla VAE or Mixture of Gaussian Prior VAE where we have access to the exact prior value). When there is an EBM in the prior distribution of the VAE, it is not possible to report "pure" ELBO because one still needs to estimate the normalization constant :
>
> $$ ELBO(\theta, \phi, \gamma) = E_{q_{\gamma}(z|x)}\left[\log p_{\phi}(x|z)\right] + E_{q_{\gamma}(z|x)} \left[\log{\frac{d(z)}{q_{\gamma}(z|x)}}\right] + E_{q_{\gamma}(z|x)}\left[-E_{\theta}(z)\right] - \log(Z) $$
> $$ = E_{q_{\gamma}(z|x)}\left[\log p_{\phi}(x|z)\right] + E_{q_{\gamma}(z|x)} \left[\log{\frac{d(z)}{q_{\gamma}(z|x)}}\right]
>     + E_{q_{\gamma}(z|x)}\left[-E_{\theta}(z)\right] - \log \int_z e^{-E_{\theta}(z)} .$$
>
> It is then possible, to estimate an upper bound of the ELBO, using yet another proposal distribution $q$ and Jensen's inequality :
>
> $$ELBO(\theta, \phi, \gamma) \leq  E_{q_{\gamma}(z|x)}\left[\log p_{\phi}(x|z)\right] + E_{q_{\gamma}(z|x)} \left[\log{\frac{d(z)}{q_{\gamma}(z|x)}}\right]
>     + E_{q_{\gamma}(z|x)}\left[-E_{\theta}(z)\right] - E_q \left[-E_{\theta}(z) \right] .$$
>
> This is not a good estimation because there is no guarantee that this is a lower bound of the likelihood $\ell_{\theta, \phi, \gamma}$. This is why we resort to SNELBO which is a lower bound of the ELBO and thus a lower bound of the likelihood. Note that this was not possible to get such an estimate of the likelihood for VAE-EBM models before SNL, maybe this is the reason why there is no ELBO or likelihood evaluation in [1]
>
> > However, this doesn't necessarily imply that the EBM trained with the techniques proposed in this paper (loss) is superior to one trained with the standard maximum likelihood loss, as practiced in previous studies[1].
>
> We want to point out to the reviewer that [1] is not doing maximum likelihood but moment matching. Indeed, [1] extend short term non persistent non convergent mcmc methods [2] from EBM to VAE-EBM. In [2], it is explicitely stated that this is not pure maximum likelihood. To do maximum likelihood, we will do longer mcmc chains than [1].
>
> > Instead, we use the plain maximum likelihood loss as shown in eq 3 and 4. (As done in [1].)
>
>
> We provide here results on Binary MNIST with two extra results. In both cases, we learn a similar model to [1] using either short term MCMC (ie 50 steps of Langevin Dynamics with the same step-size as [1]) and long term MCMC (ie 1000 steps of Langevin Dynamics using the same step size as [1]) akin to "plain maximum likelihood". We train the Short-Term MCMC VAE-EBM for 100000 training steps as [1] and the long term MCMC VAE-EBM for 25000 training steps (training is still on-going and may perform better than VAE-EBM SNELBO but is much slower).
>
> |                             |  (SN)ELBO  |  FID  |
> | --------------------------- |:----------:|:-----:|
> | VAE                         |   -89.10   | 17.23 |
> | VAE-MoG                     |   -88.73   | 14.3  |
> | {VAE-EBM SNELBO Post-Hoc}   |   -88.11   | 14.24 |
> | VAE-EBM SNELBO              | **-87.09** | 6.42  |
> | Short Term MCMC VAE-EBM [1] |    -850.1    | 21.68 |
> | Long Term MCMC VAE-EBM      |    -980.3   | 8.06  |
>
>
>
> Estimating SNELBO with MCMC is not straightforward. First, there is a lot of variance in the normalization constant estimate required calculating the SNELBO. More over, SNELBO requires an approximate proposal. Here we train a posteriori (i.e., after training) an approximate posterior with a Gaussian parameterized with the same Neural network. This is the same setting as the other VAE, but the bad results in ELBO may show that this is not a correct parameterization for the posterior. We also report the FID and show that VAE-EBM (trained with SNELBO) performed better.
>
> [1] Pang, B., Han, T., Nijkamp, E., Zhu, S. C., & Wu, Y. N. (2020). Learning latent space energy-based prior model. Advances in Neural Information Processing Systems, 33, 21994-22008.
>
> [2] Nijkamp, E., Hill, M., Zhu, S. C., & Wu, Y. N. (2019). Learning non-convergent non-persistent short-run MCMC toward energy-based model. Advances in Neural Information Processing Systems, 32.

---

### Author Response · Authors · 2023-11-19

We want to thank all reviewers for their valuable comments. All reviewers found the paper interesting, well written and the formulation of the new loss elegant.


All reviewers pointed out the need for large-scale experiments such as Image Modeling. Though we acknowledge that our current experiments remain relatively low dimensional, we think they are valuable and show the benefit of our method. To the best of our knowledge, we are the first to train an EBM for density estimation on the UCI datasets (and actually provide a good interval estimate of the likelihood). We compare to more complicated methods like Flows and perform on par while using simpler neural networks and a simple Gaussian proposal. We also provide state-of-the-art experiments on EBMs for regression.
We politely bring to the attention of the reviewers that the focus of this work is the novel method to train EBM that is grounded in theory, and for which we show its efficiency. We believe that scaling the method is another avenue of research that we leave for future work.

---

### Meta-Review · Area_Chair_RQV6 · 2023-12-08

**Metareview:**

This paper introduces a new approach coined d self-normalised log-likelihood (SNL) for training energy-based models (EBMs) that does not require expensive Markov chain Monte Carlo (MCMC) simulations, as is standard in host of competing methods -- instead, the method relies on importance sampling. The method introduces a single additional learnable parameter that represents the normalisation constant (in the spirit of NCE). The paper shows that the SNL objective is concave in the model parameters for exponential family distributions and can easily be optimized. The mehod is validated through various low-dimensional density estimation tasks and regression.

**Justification For Why Not Higher Score:**

EBM are very often used for modelling high-dimensional probability distributions -- it is crucial that the authors investigate the behaviour of the proposed method in this regime.

**Justification For Why Not Lower Score:**

N/A.

---

### Decision · Program_Chairs · 2024-01-16

Reject